# FEDERATED LEARNING BASED ON DYNAMIC REGULARIZATION

**Durmus Alp Emre Acar**[*]
alpacar@bu.edu

**Yue Zhao**[†]
yue.zhao@arm.com

**Ramon Matas Navarro**[†]
ramon.matas@arm.com

**Matthew Mattina**[†]
matthew.mattina@arm.com

**Paul N. Whatmough**[†]
paul.whatmough@arm.com

**Venkatesh Saligrama**[*]
srv@bu.edu

## ABSTRACT

We propose a novel federated learning method for distributively training neural network models, where the server orchestrates cooperation between a subset of randomly chosen devices in each round. We view Federated Learning problem primarily from a communication perspective and allow more device level computations to save transmission costs. We point out a fundamental dilemma, in that the minima of the local-device level empirical loss are inconsistent with those of the global empirical loss. Different from recent prior works, that either attempt inexact minimization or utilize devices for parallelizing gradient computation, we propose a dynamic regularizer for each device at each round, so that in the limit the global and device solutions are aligned. We demonstrate both through empirical results on real and synthetic data as well as analytical results that our scheme leads to efficient training, in both convex and non-convex settings, while being fully agnostic to device heterogeneity and robust to large number of devices, partial participation and unbalanced data.

## 1 INTRODUCTION

In (McMahan et al., 2017), the authors proposed federated learning (FL), a concept that leverages data spread across many devices, to learn classification tasks distributively without recourse to data sharing. The authors identified four principle characteristics of FL based on several use cases. First, the *communication links* between the server and devices are unreliable, and at any time, there may only be a small subset of devices that are active. Second, data is *massively distributed*, namely the number of devices are large, while amount of data per device is small. Third, device data is *heterogeneous*, in that data in different devices are sampled from different parts of the sample space. Finally, data is *unbalanced*, in that the amount of data per device is highly variable.

The basic FL problem can be cast as one of empirical minimization of a global loss objective, which is decomposable as a sum of device-level empirical loss objectives. The number of communication rounds, along with the amount of bits communicated per round, has emerged as a fundamental *gold standard* for FL problems. Many mobile and IoT devices are bandwidth constrained, and wireless transmission and reception is significantly more power hungry than computation (Halgamuge et al., 2009). As such schemes that reduce communication are warranted. While distributed SGD is a viable method in this context, it is nevertheless communication inefficient.

*A Fundamental Dilemma.* Motivated by these ideas, recent work has proposed to push optimization burden onto the devices, in order to minimize amount of communications. Much of the work in this context, propose to optimize the local risk objective based on running SGD over mini-batched device data, analogous to what one would do in a centralized scenario. On the one hand, training models on local data that minimize local empirical loss appears to be meaningful, but yet, doing so,

---

[*]Boston University, Boston, MA
[†]Arm ML Research Lab, Boston, MA

is fundamentally *inconsistent* with minimizing the global empirical loss[1] (Malinovsky et al., 2020; Khaled et al., 2020a) . Prior works (McMahan et al., 2017; Karimireddy et al., 2019; Reddi et al., 2020) attempt to overcome this issue by running fewer epochs or rounds of SGD on the devices, or attempt to stabilize server-side updates so that the resulting fused models correspond to inexact minimizations and can result in globally desirable properties.

*Dynamic Regularization.* To overcome these issues, we revisit the FL problem, and view it primarily from a communication perspective, with the goal of minimizing communication, and as such allowing for significantly more processing and optimization at the device level , since communication is the main source of energy consumption (Yadav & Yadav, 2016; Latré et al., 2011) . This approach, while increasing computation for devices, leads to substantial improvement in communication efficiency over existing state-of-the-art methods, uniformly across the four FL scenarios (unreliable links, massive distribution, substantial heterogeneity, and unbalanced data). Specifically, in each round, we dynamically modify the device objective with a penalty term so that, in the limit, when model parameters converge, they do so to stationary points of the global empirical loss. Concretely, we add linear and quadratic penalty terms, whose minima is consistent with the global stationary point. We then provide an analysis of our proposed FL algorithm and demonstrate convergence of the local device models to models that satisfy conditions for local minima of global empirical loss with a rate of $O\left(\frac{1}{T}\right)$ where $T$ is number of rounds communicated. For convex smooth functions, with $m$ devices, and $P$ devices active per round, our convergence rate for average loss with balanced data scales as $O\left(\frac{1}{T}\sqrt{\frac{m}{P}}\right)$, substantially improving over the state-of-art (SCAFFOLD $O\left(\frac{1}{T}\frac{m}{P}\right)$). For non-convex smooth functions, we establish a rate of $O\left(\frac{1}{T}\frac{m}{P}\right)$.

We perform experiments on both visual and language real-world datasets including MNIST, EMNIST, CIFAR-10, CIFAR-100 and Shakespeare. We tabulate performance studying cases that are reflective of FL scenarios, namely, for (i) varying device participation levels, (ii) massively distributed data, (iii) various levels of heterogeneity, as well as (iv) unbalanced local data settings. Our proposed algorithm, FedDyn, has similar overhead to competing approaches, but converges at a significantly faster rate. This results in a substantial reduction in communication compared to baseline approaches such as conventional FedAvg (McMahan et al., 2017), FedProx (Li et al., 2020) and SCAFFOLD (Karimireddy et al., 2019), for achieving target accuracy. Furthermore, our approach is simple to implement, requiring far less hyperparameter tuning compared to competing methods.

**Contributions.** We summarize our main results here.

- We present, FedDyn, a novel dynamic regularization method for FL. Key to FedDyn is a new concept, where in each round the risk objective for each device is dynamically updated so as to ensure that the device optima is asymptotically consistent with stationary points of the global empirical loss,
- We prove convergence results for FedDyn in both convex and non-convex settings, and obtain sharp results for communication rounds required for achieving target accuracy. Our results for convex case improves significantly over state-of-art prior works. FedDyn in theory is unaffected by heterogeneity, massively distributed data, and quality of communication links,
- On benchmark examples FedDyn achieves significant communication savings over competing methods uniformly across various choices of device heterogeneity and device participation on massively distributed large-scale text and visual datasets.

**Related Work.** FL is a fast evolving topic, and we only describe closely related approaches here. Comprehensive field studies have appeared in (Kairouz et al., 2019; Li et al., 2020). The general FL setup involves two types of updates, the server and device, and each of these updates are associated with minimizing some local loss function, which by itself could be updated dynamically over different rounds. At any round, there are methods that attempt to fully optimize or others that propose inexact optimization. We specifically focus on relevant works that address the four FL scenarios (massive distribution, heterogeneity, unreliable links, and unbalanced data) here.

---

[1]To see this consider the situation where losses are differentiable. As such stationary points for global empirical loss demand that only the sum of the gradients of device empirical losses are zero, and not necessarily that the individual device gradients are zero. Indeed, in statistically heterogeneous situations, such as where we have heterogeneous dominance of classes, stationary points of local empirical functions do not coincide.

One line of work proposes local SGD (Stich, 2019) based updates, wherein each participating device performs a single local SGD step. The server then averages received models. In contrast to local SGD, our method proposes to minimize a local penalized empirical loss.

FedAvg (McMahan et al., 2017) is a generalization of local SGD, which proposes a larger number of local SGD steps per round. Still, FedAvg inexactly solves device side optimization. Identifying when to stop minimizing so that one gets a good accuracy-communication trade-off is based on tuning the number of epochs and the learning rate (McMahan et al., 2017; Li et al., 2020b). Despite the strong empirical performance of FedAvg in IID settings, performance degrades in non-IID scenarios (Zhao et al., 2018).

Several modifications of FedAvg have been proposed to handle non-IID settings. These variants include using a decreasing learning rate (Li et al., 2020b); modifying device empirical loss dynamically (Li et al., 2020a); or modifying server side updates (Hsu et al., 2019; Reddi et al., 2020). Methods that use a decreasing learning rate or customized server side updates still rely on local SGD updates within devices. While these works do recognize the incompatibility of local and global stationary points, their proposed fix is based on inexact minimization. Additionally, in order to establish convergence for non-IID situations, these works impose additional "bounded-non-IID" conditions.

FedProx (Li et al., 2020a) is related to our method. Like us they propose a dynamic regularizer, which is modified based on server supplied models. This regularizer penalizes updates that are far away from the server model. Nevertheless, the resulting regularizer does not result in aligning the global and local stationary points, and as such inexact minimization is warranted, and they do so by carefully choosing learning rates and epochs. Furthermore, tuning requires some knowledge of statistical heterogeneity.

In a similar vein, there are works that augment updates with extra device variables that are also transmitted along with the models (Karimireddy et al., 2019; Shamir et al., 2014). These works prove convergence guarantees through adding device-dependent regularizers. Nevertheless, they suffer additional communication costs and they are not extensively experimented with deep neural networks. Among them, SCAFFOLD (Karimireddy et al., 2019) is a closely related work even though it transmits extra variables and a more detailed comparison is given in Section 2.

Another line of distributed optimization methods (Konečnỳ et al., 2016; Makhdoumi & Ozdaglar, 2017; Shamir et al., 2014; Yuan & Ma, 2020; Pathak & Wainwright, 2020; Liang et al., 2019; Li et al., 2020c; Condat et al., 2020) could be considered in this setting. Moreover, there are works that extend analysis of SGD type methods to FL settings (Gorbunov et al., 2020; Khaled et al., 2020b; Li & Richtárik, 2020). However, these algorithms are proposed for full device participation case which fails to satisfy one important aspect of FL. FedSVRG (Konečnỳ et al., 2016) and DANE (Shamir et al., 2014) need gradient information from all devices at each round and they are not directly applicable to partial FL settings. For example, FedDANE (Li et al., 2019) is a version of DANE that works in partial participation. However, FedDANE performs worse than FedAvg empirically with partial participation (Li et al., 2019). Similar to these works, FedPD (Zhang et al., 2020) method is proposed in distributed optimization with a different participation notion. FedPD activates either all devices or no devices per round which again fails to satisfy partial participation in FL.

Lastly, another set of works aims to decrease communication costs by compressing the transmitted models (Dutta et al., 2019; Mishchenko et al., 2019; Alistarh et al., 2017). They save communication costs through decreasing bit-rate of the transmission. These ideas are complementary to our work and they can be integrated to our proposed solution.

## 2 METHOD

We assume there is a cloud server which can transmit and receive messages from $m$ client devices. Each device, $k \in [m]$ consists of $N_k$ training instances in the form of features, $\boldsymbol{x} \in \mathcal{X}$ and corresponding labels $y \in \mathcal{Y}$ that are drawn IID from a device-indexed joint distribution, $(\boldsymbol{x}, y) \sim P_k$.

Our objective is to solve

$$\arg \min_{\boldsymbol{\theta} \in \mathbb{R}^d} \left[ \ell(\boldsymbol{\theta}) \triangleq \frac{1}{m} \sum_{k \in [m]} L_k(\boldsymbol{\theta}) \right]$$

---

**Algorithm 1:** Federated Dynamic Regularizer - (FedDyn)

---

**Input:** $T, \boldsymbol{\theta}^0, \alpha > 0, \nabla L_k(\boldsymbol{\theta}_k^0) = \mathbf{0}$.
**for** $t = 1, 2, \ldots T$ **do**

    Sample devices $\mathcal{P}_t \subseteq [m]$ and transmit $\boldsymbol{\theta}^{t-1}$ to each selected device,

    **for** *each device $k \in \mathcal{P}_t$, and in parallel* **do**

        Set $\boldsymbol{\theta}_k^t = \arg\min_{\boldsymbol{\theta}} L_k(\boldsymbol{\theta}) - \langle \nabla L_k(\boldsymbol{\theta}_k^{t-1}), \boldsymbol{\theta} \rangle + \frac{\alpha}{2} \|\boldsymbol{\theta} - \boldsymbol{\theta}^{t-1}\|^2$,

        Set $\nabla L_k(\boldsymbol{\theta}_k^t) = \nabla L_k(\boldsymbol{\theta}_k^{t-1}) - \alpha \left( \boldsymbol{\theta}_k^t - \boldsymbol{\theta}^{t-1} \right)$,

        Transmit device model $\boldsymbol{\theta}_k^t$ to server,

    **end for**

    **for** *each device $k \notin \mathcal{P}_t$, and in parallel* **do**

        Set $\boldsymbol{\theta}_k^t = \boldsymbol{\theta}_k^{t-1}, \nabla L_k(\boldsymbol{\theta}_k^t) = \nabla L_k(\boldsymbol{\theta}_k^{t-1})$,

    **end for**

    Set $\boldsymbol{h}^t = \boldsymbol{h}^{t-1} - \alpha \frac{1}{m} \left( \sum_{k \in \mathcal{P}_t} \boldsymbol{\theta}_k^t - \boldsymbol{\theta}^{t-1} \right)$,

    Set $\boldsymbol{\theta}^t = \left( \frac{1}{|\mathcal{P}_t|} \sum_{k \in \mathcal{P}_t} \boldsymbol{\theta}_k^t \right) - \frac{1}{\alpha} \boldsymbol{h}^t$

**end for**

---

where, $L_k(\boldsymbol{\theta}) = \mathbb{E}_{(\boldsymbol{x},y)\sim\mathcal{D}_k}[\ell_k(\boldsymbol{\theta}; (\boldsymbol{x}, y))]$ is the empirical loss of the $k$th device, and $\boldsymbol{\theta}$ are the parameters of our neural network, whose structure is assumed to be identical across the devices and the server. We denote by $\boldsymbol{\theta}^*$ a local minima of the global empirical loss function.

**FedDyn Method.** Our proposed method, FedDyn, is displayed in Algorithm 1. In each round, $t \in [T]$, a subset of devices $\mathcal{P}_t \subset [m]$ are active, and the server transmits its current model, $\boldsymbol{\theta}^{t-1}$, to these devices. Each active device then optimizes a local empirical risk objective, which is the sum of its local empirical loss and a penalized risk function. The penalized risk, which is dynamically updated, is based on current local device model, and the received server model:

$$\boldsymbol{\theta}_k^t = \arg\min_{\boldsymbol{\theta}} \left[ \mathfrak{R}_k(\boldsymbol{\theta}; \boldsymbol{\theta}_k^{t-1}, \boldsymbol{\theta}^{t-1}) \triangleq L_k(\boldsymbol{\theta}) - \langle \nabla L_k(\boldsymbol{\theta}_k^{t-1}), \boldsymbol{\theta} \rangle + \frac{\alpha}{2} \|\boldsymbol{\theta} - \boldsymbol{\theta}^{t-1}\|^2 \right]. \quad (1)$$

Devices compute their local gradient, $\nabla L_k \left( \boldsymbol{\theta}_k^{t-1} \right)$, recursively, by noting that the first order condition for local optima must satisfy,

$$\nabla L_k(\boldsymbol{\theta}_k^t) - \nabla L_k(\boldsymbol{\theta}_k^{t-1}) + \alpha(\boldsymbol{\theta}_k^t - \boldsymbol{\theta}^{t-1}) = \mathbf{0} \quad (2)$$

Stale devices do not update their models. Updated device models, $\boldsymbol{\theta}_k^t$, $k \in \mathcal{P}_t$ are then transmitted to server, which then updates its model to $\boldsymbol{\theta}^t$ as displayed in Algorithm 1.

*Intuitive Justification.* To build intuition into our method, we first highlight a fundamental issue about the Federated Dynamic Regularizer setup. It is that stationary points for device losses, in general, do not conform to global losses. Indeed, a global stationary point, $\boldsymbol{\theta}^*$ must necessarily satisfy,

$$\nabla \ell(\boldsymbol{\theta}^t) \triangleq \frac{1}{m} \sum_{k \in [m]} \nabla L_k(\boldsymbol{\theta}_*) = \sum_{k \in [m]} \mathbb{E}_{(\boldsymbol{x},y)\sim\mathcal{D}_k} \nabla \ell_k(\boldsymbol{\theta}_*; (\boldsymbol{x}, y)) = \mathbf{0}. \quad (3)$$

In contrast a device's stationary point, $\boldsymbol{\theta}_k^*$ satisfies, $\nabla L_k(\boldsymbol{\theta}_k^*) = \mathbf{0}$, and in general due to heterogeneity of data ($P_k \neq P_j$ for $k \neq j$), the individual device-wise gradients are non-zero $\nabla L_k(\boldsymbol{\theta}_*) \neq \mathbf{0}$. This means that the dual goals of (i) seeking model convergence to a consensus, namely, $\boldsymbol{\theta}_k^t \to \boldsymbol{\theta}^t \to \boldsymbol{\theta}_*$, and (ii) the fact that model updates are based on optimizing local empirical losses is inconsistent[2].

**Dynamic Regularization.** Our proposed risk objective in Eq. 1 dynamically modifies local loss functions, so that, if in fact local models converge to a consensus, the consensus point is consistent with stationary point of the global loss. To see this, first note that if we initialize at a consensus point, namely, $\boldsymbol{\theta}_k^{t-1} = \boldsymbol{\theta}^{t-1}$, we have, $\nabla \mathfrak{R}(\boldsymbol{\theta}, \boldsymbol{\theta}_k^{t-1}, \boldsymbol{\theta}^{t-1}) = \mathbf{0}$ for $\boldsymbol{\theta} = \boldsymbol{\theta}^{t-1}$. Thus our choice can be

---

[2]As pointed in related work prior works based on SGD implicitly account for the inconsistency by performing inexact minimization, and additional hyperparameter tuning.

seen as modifying the device loss so that the stationary points of device risk is consistent with server model.

*Key Property of Algorithm 1.* If local device models converge, they converge to the server model, and the convergence point is a stationary point of the global loss. To see this, observe from Eq 2 that if $\boldsymbol{\theta}_k^t \to \boldsymbol{\theta}_k^\infty$, it generally follows that, $\nabla L_k(\boldsymbol{\theta}_k^t) \to \nabla L_k(\boldsymbol{\theta}_k^\infty)$, and as a consequence, we have $\boldsymbol{\theta}^t \to \boldsymbol{\theta}_k^\infty$. In turn this implies that $\boldsymbol{\theta}_k^\infty \to \boldsymbol{\theta}^\infty$ , i.e., is independent of $k$. Putting all of this together with our server update equations we have that $\boldsymbol{\theta}^t$ convergence implies $\boldsymbol{h}^t \to 0$. Now the server state $\boldsymbol{h}^t \triangleq \sum_k \nabla L_k(\boldsymbol{\theta}_k^t)$, and as such in the limit we are left with $\sum_k \nabla L_k(\boldsymbol{\theta}_k^t) \to \sum_k \nabla L_k(\boldsymbol{\theta}^\infty) = \mathbf{0}$. This implies that we converge to a point that turns out to be a stationary point of the global risk.

## 2.1 CONVERGENCE ANALYSIS OF FEDDYN.

Properties outlined in the previous section, motivates our FedDyn convergence analysis of device and server models. We will present theoretical results for strongly convex, convex and non-convex functions.

**Theorem 1.** *Assuming a constant number of devices are selected uniformly at random in each round, $|\mathcal{P}_t| = P$, for a suitably chosen of $\alpha > 0$, Algorithm 1 satisfies,*

- *$\mu$ strongly convex and $L$ smooth $\{L_k\}_{k=1}^m$ functions,*

$$E\left[\ell\left(\frac{1}{R}\sum_{t=0}^{T-1} r^t \boldsymbol{\gamma}^t\right) - \ell_*\right] = O\left(\frac{1}{r^T}\left(\beta \left\|\boldsymbol{\theta}^0 - \boldsymbol{\theta}_*\right\|^2 + \frac{m}{P}\frac{1}{\beta}\left(\frac{1}{m}\sum_{k\in[m]}\left\|\nabla L_k(\boldsymbol{\theta}_*)\right\|^2\right)\right)\right)$$

- *Convex and $L$ smooth $\{L_k\}_{k=1}^m$ functions,*

$$E\left[\ell\left(\frac{1}{T}\sum_{t=0}^{T-1}\boldsymbol{\gamma}^t\right) - \ell_*\right] = O\left(\frac{1}{T}\sqrt{\frac{m}{P}}\left(L\left\|\boldsymbol{\theta}^0 - \boldsymbol{\theta}_*\right\|^2 + \frac{1}{L}\frac{1}{m}\sum_{k\in[m]}\left\|\nabla L_k(\boldsymbol{\theta}_*)\right\|^2\right)\right)$$

- *Nonconvex and $L$ smooth $\{L_k\}_{k=1}^m$ functions,*

$$E\left\|\nabla\ell(\overline{\boldsymbol{\gamma}}_T)\right\|^2 = O\left(\frac{1}{T}\left(L\frac{m}{P}\left(\ell(\boldsymbol{\theta}^0) - \ell_*\right) + L^2\frac{1}{m}\sum_{k\in[m]}\left\|\boldsymbol{\theta}_k^0 - \boldsymbol{\theta}^0\right\|^2\right)\right)$$

*where $\boldsymbol{\gamma}^t = \frac{1}{P}\sum_{k\in\mathcal{P}_t}\boldsymbol{\theta}_k^t$, $\boldsymbol{\theta}_* = \arg\min_{\boldsymbol{\theta}}\ell(\boldsymbol{\theta})$, $\ell_* = \ell(\boldsymbol{\theta}_*)$ , $r = \left(1 + \frac{\mu}{\alpha}\right)$, $R = \sum_{t=0}^{T-1} r^t$ $\beta = \max\left(5\frac{m}{P}\mu, 30L\right)$ and $\overline{\boldsymbol{\gamma}}_T$ is a random variable that takes values $\{\boldsymbol{\gamma}^s\}_{s=0}^{T-1}$ with equal probability.*

Theorem 1 gives rates for strongly convex, convex and nonconvex local losses. For strongly convex and smooth functions, in expectation, a weighted average of active device averages converge at a linear rate. For convex and smooth functions, in expectation, the global loss of active device averages, converges at a rate $O\left(\frac{1}{T}\sqrt{\frac{m}{P}}\right)$. Following convention, this rate is for the empirical loss averaged across devices. As such this rate would hold with moderate data imbalance. In situations with significant imbalance, which scales with data size, these results would have to account for the variance in the amount of data/device. Furthermore, the $\sqrt{\frac{m}{P}}$ factor might appear surprising, but note that our bounds hold under expectation, namely, the error reflects the average over all random choices of devices. Similarly, for nonconvex and smooth functions, in expectation, average of active device models converges to a stationary point at $O\left(\frac{1}{T}\frac{m}{P}\right)$ rate. The expectation is taken over randomness in active device set at each round. Similar to known convergence theorems, the problem dependent constants are related to how good the algorithm is initialized. We refer to Appendix B for a detailed proof.

**FedDyn vs. SCAFFOLD (Karimireddy et al., 2019).** While SCAFFOLD appears to be similar to our method, there are fundamental differences. Practically, SCAFFOLD communicates twice as many bits as FedDyn or Federated Dynamic Regularizer, transmitting back and forth, both

Table 1: Number of parameters transmitted relative to one round of FedAvg to reach target test accuracy for moderate and large number of devices in IID and Dirichlet .3 settings. SCAFFOLD communicates the current model and its associated gradient per round, while others communicate only the current model. As such number of rounds for SCAFFOLD is one half of those reported.

| Device Number | Dataset | Accuracy | FedDyn | SCAFFOLD | FedAvg | FedProx |
|---|---|---|---|---|---|---|
| **Moderate** | | | | **IID** | | |
| | CIFAR-10 | 84.5 | 637 | 1852(2.9×) | 1000+(>1.6×) | 1000+(>1.6×) |
| | | 82.3 | 240 | 512(2.1×) | 994(4.1×) | 825(3.4×) |
| | CIFAR-100 | 51.0 | 522 | 1854(3.6×) | 1000+(>1.9×) | 1000+(>1.9×) |
| | | 40.9 | 159 | 286(1.8×) | 822(5.2×) | 873(5.5×) |
| | | | | **Dirichlet (.3)** | | |
| | CIFAR-10 | 82.5 | 444 | 1880(4.2×) | 1000+(>2.3×) | 1000+(>2.3×) |
| | | 80.7 | 232 | 594(2.6×) | 863(3.7×) | 930(4.0×) |
| | CIFAR-100 | 51.0 | 561 | 1884(3.4×) | 1000+(>1.8×) | 1000+(>1.8×) |
| | | 42.3 | 170 | 330(1.9×) | 959(5.6×) | 882(5.2×) |
| **Massive** | | | | **IID** | | |
| | CIFAR-10 | 80.0 | 840 | 4000+(>4.8×) | 2000+(>2.4×) | 2000+(>2.4×) |
| | | 62.3 | 305 | 928(3.0×) | 1277(4.2×) | 1274(4.2×) |
| | CIFAR-100 | 50.1 | 1445 | 3982(2.8×) | 2000+(>1.4×) | 2000+(>1.4×) |
| | | 38.3 | 477 | 1408(3.0×) | 1997(4.2×) | 1974(4.1×) |
| | | | | **Dirichlet (.3)** | | |
| | CIFAR-10 | 80.0 | 831 | 4000+(>4.8×) | 2000+(>2.4×) | 2000+(>2.4×) |
| | | 70.6 | 350 | 2138(6.1×) | 1962(5.6×) | 1517(4.3×) |
| | CIFAR-100 | 47.0 | 969 | 4000(4.1×) | 2000+(>2.1×) | 2000+(>2.1×) |
| | | 39.9 | 467 | 2266(4.9×) | 1913(4.1×) | 1794(3.8×) |

a model and its gradient. The $2\times$ increase in bits can be substantial for many low-power IoT applications, since energy consumption for communication dominates computation. Conceptually, we attribute the increased bit-rate to algorithmic differences. At the device-level, our modified risk incorporates a linear term, $\nabla L_k(\boldsymbol{\theta}_k^t)$ (which we can compute readily (Eq. 2)). Applying our perspective to SCAFFOLD, in full participation setting, we see SCAFFOLD as replacing our linear term $\langle \nabla L_k(\boldsymbol{\theta}_k^t), \boldsymbol{\theta} \rangle$ with $\langle \nabla L_k(\boldsymbol{\theta}^t) - \frac{1}{m} \sum_{k \in [m]} \nabla L_k(\boldsymbol{\theta}_k^t), \boldsymbol{\theta} \rangle$. While $\nabla L_k(\boldsymbol{\theta}^t)$ can be locally computed, after $\boldsymbol{\theta}^t$ is received, the term $\frac{1}{m} \sum_{k \in [m]} \nabla L_k(\boldsymbol{\theta}_k^t)$ is unknown and must be transmitted by the server, leading to increased bit-rate. Note that this is unavoidable, since ignoring this term, leads to freezing device updates (optimizing $L_k(\boldsymbol{\theta}) - \langle \nabla L_k(\boldsymbol{\theta}^t), \boldsymbol{\theta} - \boldsymbol{\theta}^t \rangle + \frac{\alpha}{2} \|\boldsymbol{\theta} - \boldsymbol{\theta}^t\|^2$ results in $\boldsymbol{\theta} = \boldsymbol{\theta}^t$). This extra term is a surrogate for $\nabla \ell(\boldsymbol{\theta}^t)$, which is unavailable. As such we believe that these differences are responsible for FedDyn's improved rate (in rounds) in theory as well as practice.

Finally, apart from conceptual differences, there are also implementation differences. SCAFFOLD runs SGD, and adapts hyperparameter tuning for a given number of rounds to maximize accuracy. In contrast, our approach, based on exact minimization, is agnostic to specific implementation, and as such we utilize significantly less tuning.

## 3 Experiments

Our goal in this section is to evaluate FedDyn against competing methods on benchmark datasets for various FL scenarios. Our results will highlight tradeoffs and benefits of our exact minimization relative to prior inexact minimization methods. To ensure a fair comparison, the usual SGD procedure is adapted for the FedDyn algorithm in the device update as in FedAvg rather than leveraging an off the shelf optimization solver. We provide a brief description of the datasets and the models used in the experiments. A detailed description of our setup can be found in Appendix A.1. Partial participation was handled by sampling devices at random in each round independent of previous rounds.

**Datasets.** We used benchmark datasets with the same train/test splits as in previous works (McMahan et al., 2017; Li et al., 2020a) which are MNIST (LeCun et al., 1998), CIFAR-10, CIFAR-100 (Krizhevsky et al., 2009), a subset of EMNIST (Cohen et al., 2017) (EMNIST-L), Shakespeare (Shakespeare, 1994) as well as a synthetic dataset. The IID split is generated by randomly assigning datapoints to the devices. The Dirichlet distribution is used on the label ratios to ensure uneven label distributions among devices for non-IID splits as in (Yurochkin et al., 2019). For example, in MNIST, 100 device experiments, each device has about 5 and 3 classes that consume 80% of local

data at Dirichlet parameter settings of 0.6 and 0.3 respectively. To generate unbalanced data, we sample the number of datapoints from a lognormal distribution. Controlling the variance of lognormal distribution gives unbalanced data. For instance, in CIFAR-10, 100 device experiments, balanced and unbalanced data settings have standard deviation of device sample size of 0 and 0.3 respectively.

**Models.** We use fully-connected neural network architectures for MNIST and EMNIST-L with 2 hidden layers. The number of neurons in the layers are 200 and 100; and the models achieve $98.4\%$ and $95.0\%$ test accuracy in MNIST and EMNIST-L respectively. The model used for MNIST is the same as used in (McMahan et al., 2017). For CIFAR-10 and CIFAR-100, we use a CNN model, similar to (McMahan et al., 2017), consisting of 2 convolutional layers with $64\ 5 \times 5$ filters followed by 2 fully connected layers with 394 and 192 neurons, and a softmax layer. The model achieves $85.2\%$ and $55.3\%$ test accuracy for CIFAR-10 and CIFAR-100 respectively. For the next character prediction task (Shakespeare), we use a stacked LSTM, similar to (Li et al., 2020a). This architecture achieves a test accuracy of $50.8\%$ and $51.2\%$ in IID and non-IID settings respectively. Both IID and non-IID performances are reported since splits are randomly regenerated from the entire Shakespeare writing. Hence centralized data and the centralized model performance is different.

In passing, we note that while the accuracies reported are state-of-art for our chosen models, higher capacity models can achieve higher performance on these datasets. As such, our aim is to compare the relative performance of these models in FL using FedDyn and other strong baselines.

**Comparison of Methods.** We report the performance of FedDyn, SCAFFOLD, FedAvg and FedProx on synthetic and real datasets. We also experimented with distributed SGD, where devices in each round compute gradients on the server supplied model on local data, and communicate these gradients. Its performance was not competitive relative to other methods. Therefore, we do not tabulate it here. We cover synthetic data generation and its results in Appendix A.1.

The standard goal in FL is to minimize amount of bits transferred. For this reason, we adopt the number of models transmitted to achieve a target accuracy as our metric in our comparisons. This metric is different than comparing communication rounds since not all methods communicate the same amount of information per round. FedDyn, FedAvg and FedProx transmit/receive the same amount of models for a fixed number of rounds whereas SCAFFOLD costs twice due to transmission of states. We compare algorithms for two different accuracy levels which we pick them to be close to performance obtained by centralizing data. Along with transmission costs of each method, we report the communication savings of FedDyn compared to each baseline in parenthesis. For methods that could not achieve aimed accuracy within the communication constraint, we append transmission cost with $+$ sign. We observe FedDyn results in communication savings compared to all baselines to reach a target accuracy. We test FedDyn under the four characteristic properties of FL which are partial participation, large number of devices, heterogeneous data, and unbalanced data.

**Moderate vs. Large Number of Devices.** *FedDyn significantly outperforms competing methods in the practically relevant massively distributed scenario.* We report the performance of FedDyn on CIFAR-10 and CIFAR-100 with moderate and large number of devices in Table 1, while keeping the participation level constant $(10\%)$ and the data amounts balanced. Specifically, the moderately distributed setting has 100 devices with 500 images per device. The massively distributed setting has 1000 devices with 50 images per device for CIFAR-10, as well as 500 devices with 100 images per device for CIFAR-100. In each distributed setting, the data is partitioned in both IID and non-IID (Dirichlet 0.3) fashion. FedDyn leads to substantial transmission reduction in each of the regimes.

First, the communication saving in the massive setting is significantly larger relative to the moderate setting. Compared to SCAFFOLD, FedDyn leads to $4.8\times$ and $2.9\times$ gains respectively on CIFAR-10 IID setting. SCAFFOLD is not able to achieve $80\%$ within 2000 rounds in the massive setting (shown in Figure 4a), thus actual saving is more than $4.8\times$. Similar trend is observed in the non-IID setting of CIFAR-10 and CIFAR-100. Second, all the methods require more communications to achieve a reasonable accuracy in the massive setting as the dataset is more decentralized. For instance, it takes FedDyn 637 rounds to achieve $84.5\%$ with 100 devices, while it takes 840 rounds to achieve $80.0\%$ with 1000 devices. Similar trend is observed for CIFAR-100 and other methods. FedDyn always achieves the target accuracy with fewer rounds and thus leads to significant saving. Third, a higher target accuracy may result in a greater saving. For instance, the saving relative to SCAFFOLD increases from $3\times$ to $4.8\times$ in the CIFAR-10 IID massive setting. We may attribute this to the fact that FedDyn aligns device functions to global loss and efficiently optimizes the problem.

**Full vs. Partial Participation Levels.** *FedDyn outperforms baseline methods across different device participation levels.* We consider different device participation levels with 100 devices and balanced data in Table 2 where part of CIFAR-10 and CIFAR-100 results are omitted since they are reported in moderate number of devices section of Table 1. The Shakespeare non-IID results are separately shown, since it has a natural non-IID split which does not conform with the Dirichlet distribution. The communication gain, with respect to best baseline, increases with greater participation levels from $2.9\times$ to $9.4\times$; $4.0\times$ to $12.8\times$ and $4.2\times$ to $7.9\times$ for CIFAR-10 in different device distribution settings. We observe a similar performance increase in full participation for most of the datasets. This validates our hypothesis that FedDyn more efficiently incorporates information from all devices compared to other methods, and results in more savings in full participation. Similar to previous results, a greater target accuracy gives a greater savings in most of the settings. We also report results for $1\%$ participation regime with different device distribution settings (See Table 5 in Appendix A.1).

**Balanced vs. Unbalanced Data.** *FedDyn is more robust to unbalanced data than competing methods.* We fix number of devices (100) and participation level (10%) and consider effect of unbalanced data (Table 4 (Appendix A.1)). FedDyn achieves $4.3\times$ gains over the best competitor, SCAFFOLD to achieve the target accuracy. As before, gains increase with the target accuracy.

**IID vs. non-IID Device Distribution.** *FedDyn outperforms baseline methods across different device distribution levels.* We consider heterogeneous device distributions in the context of varying device numbers, participation levels and balanced-unbalanced settings in Table 1, 2 and 4 (Appendix A.1) respectively. Device distributions become more non-IID as we go from IID, Dirichlet .6 to Dirichlet .3 splits which makes global optimization problem harder. We see a clear effect of this change in Table 2 for $10\%$ participation level and in Table 4 for unbalanced setting. For instance, increasing non-IID level results in a greater communication saving such as from $2.9\times$, $4.0\times$ to $4.2\times$ in CIFAR-10 $10\%$ participation. Similar statement holds for MNIST, EMNIST-L and Shakespeare in Table 2 and for CIFAR-10 unbalanced setting in Table 4. We do not observe a significant difference in savings for full participation setting in Table 2.

**Summary.** Overall, FedDyn consistently leads to substantial communication savings compared to baseline methods uniformly across various FL regimes of interest. We realize large gains in the practically relevant massively distributed data setting.

## 4  CONCLUSION

We proposed FedDyn, a novel FL method for distributively training neural network models. FedDyn is based on exact minimization, wherein at each round, each participating device, dynamically updates its regularizer so that the optimal model for the regularized loss is in conformity with the global empirical loss. Our approach is different from prior works that attempt to parallelize gradient computation, and in doing so they tradeoff target accuracy with communications, and necessitate inexact minimization. We investigate different characteristic FL settings to validate our method. We demonstrate both through empirical results on real and synthetic data as well as analytical results that our scheme leads to efficient training with convergence rate as $O\left(\frac{1}{T}\right)$ where $T$ is number of rounds, in both convex and non-convex settings, and a linear rate in strongly convex setting, while being fully agnostic to device heterogeneity and robust to large number of devices, partial participation and unbalanced data.

Table 2: Number of parameters transmitted relative to one round of FedAvg to reach target test accuracy for 100% and 10% participation regimes in the IID, non-IID settings. SCAFFOLD communicates the current model and its associated gradient per round, while others communicate only the current model. As such number of rounds for SCAFFOLD is one half of those reported.

| Participation | Dataset | Accuracy | FedDyn | SCAFFOLD | FedAvg | FedProx |
|---|---|---|---|---|---|---|
| **100%** | | **IID** | | | | |
| | CIFAR-10 | 85.0 | 198 | 1860(9.4×) | 1000+(>5.1×) | 1000+(>5.1×) |
| | | 81.4 | 67 | 320(4.8×) | 754(11.3×) | 655(9.8×) |
| | CIFAR-100 | 51.0 | 259 | 1744(6.7×) | 1000+(>3.9×) | 1000+(>3.9×) |
| | | 39.4 | 55 | 172(3.1×) | 1000+(>18.2×) | 741(13.5×) |
| | MNIST | 98.2 | 38 | 72(1.9×) | 194(5.1×) | 445(11.7×) |
| | | 97.2 | 9 | 18(2.0×) | 31(3.4×) | 28(3.1×) |
| | EMNIST-L | 94.6 | 65 | 414(6.4×) | 307(4.7×) | 1000+(>15×) |
| | | 93.6 | 16 | 36(2.2×) | 66(4.1×) | 62(3.9×) |
| | Shakespeare | 46.4 | 33 | 74(2.2×) | 96(2.9×) | 113(3.4×) |
| | | 45.4 | 28 | 64(2.3×) | 59(2.1×) | 56(2.0×) |
| | | **Dirichlet (.6)** | | | | |
| | CIFAR-10 | 84.0 | 148 | 1890(12.8×) | 1000+(>6.8×) | 1000+(>6.8×) |
| | | 80.3 | 64 | 392(6.1×) | 869(13.6×) | 724(11.3×) |
| | CIFAR-100 | 51.0 | 468 | 1838(3.9×) | 1000+(>2.1×) | 1000+(>2.1×) |
| | | 40.6 | 73 | 206(2.8×) | 998(13.7×) | 592(8.1×) |
| | MNIST | 98.1 | 39 | 108(2.8×) | 157(4.0×) | 416(10.7×) |
| | | 97.1 | 11 | 24(2.2×) | 38(3.5×) | 34(3.1×) |
| | EMNIST-L | 94.9 | 207 | 552(2.7×) | 410(2.0×) | 1000+(>4.8×) |
| | | 93.9 | 20 | 42(2.1×) | 73(3.6×) | 61(3.0×) |
| | | **Dirichlet (.3)** | | | | |
| | CIFAR-10 | 83.5 | 223 | 1762(7.9×) | 1000+(>4.5×) | 1000+(>4.5×) |
| | | 80.2 | 70 | 504(7.2×) | 705(10.1×) | 1000+(>14.3×) |
| | CIFAR-100 | 50.5 | 405 | 1940(4.8×) | 1000+(>2.5×) | 1000+(>2.5×) |
| | | 41.0 | 80 | 224(2.8×) | 911(11.4×) | 1000+(>12.5×) |
| | MNIST | 98.1 | 35 | 76(2.2×) | 313(8.9×) | 458(13.1×) |
| | | 97.1 | 10 | 24(2.4×) | 49(4.9×) | 44(4.4×) |
| | EMNIST-L | 94.5 | 65 | 210(3.2×) | 492(7.6×) | 1000+(>15×) |
| | | 93.5 | 23 | 46(2.0×) | 78(3.4×) | 69(3.0×) |
| | | **Non-IID** | | | | |
| | Shakespeare | 47.3 | 33 | 70(2.1×) | 134(4.1×) | 150+(>4.5×) |
| | | 46.3 | 28 | 62(2.2×) | 53(1.9×) | 64(2.3×) |
| **10%** | | **IID** | | | | |
| | MNIST | 98.2 | 100 | 142(1.4×) | 588(5.9×) | 362(3.6×) |
| | | 97.2 | 31 | 52(1.7×) | 49(1.6×) | 43(1.4×) |
| | EMNIST-L | 94.6 | 104 | 160(1.5×) | 330(3.2×) | 210(2.0×) |
| | | 93.6 | 58 | 84(1.4×) | 69(1.2×) | 65(1.1×) |
| | Shakespeare | 46.9 | 63 | 94(1.5×) | 138(2.2×) | 190(3.0×) |
| | | 45.9 | 56 | 76(1.4×) | 96(1.7×) | 75(1.3×) |
| | | **Dirichlet (.6)** | | | | |
| | CIFAR-10 | 83.5 | 403 | 1618(4.0×) | 1000+(>2.5×) | 1000+(>2.5×) |
| | | 81.3 | 189 | 486(2.6×) | 977(5.2×) | 943(5.0×) |
| | CIFAR-100 | 51.0 | 521 | 1910(3.7×) | 1000+(>1.9×) | 1000+(>1.9×) |
| | | 41.6 | 170 | 302(1.8×) | 931(5.5×) | 748(4.4×) |
| | MNIST | 98.1 | 129 | 194(1.5×) | 581(4.5×) | 361(2.8×) |
| | | 97.1 | 37 | 60(1.6×) | 57(1.5×) | 57(1.5×) |
| | EMNIST-L | 94.9 | 192 | 296(1.5×) | 306(1.6×) | 1000+(>5.2×) |
| | | 93.9 | 55 | 102(1.9×) | 95(1.7×) | 86(1.6×) |
| | | **Dirichlet (.3)** | | | | |
| | MNIST | 98.2 | 90 | 208(2.3×) | 428(4.8×) | 858(9.5×) |
| | | 97.2 | 37 | 68(1.8×) | 76(2.1×) | 61(1.6×) |
| | EMNIST-L | 94.4 | 107 | 178(1.7×) | 804(7.5×) | 1000+(>9.3×) |
| | | 93.4 | 58 | 100(1.7×) | 81(1.4×) | 86(1.5×) |
| | | **Non-IID** | | | | |
| | Shakespeare | 47.6 | 63 | 102(1.6×) | 169(2.7×) | 133(2.1×) |
| | | 46.6 | 56 | 82(1.5×) | 80(1.4×) | 66(1.2×) |

ACKNOWLEDGEMENTS

This research was supported by a gift from ARM corporation (DA), and CCF-2007350 (VS), CCF-2022446(VS), CCF-1955981 (VS), the Data Science Faculty Fellowship from the Rafik B. Hariri Institute.

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

# A    APPENDIX

## A.1    EXPERIMENT DETAILS

### A.1.1    SYNTHETIC DATA

**Dataset.** We introduce a synthetic dataset to reflect different properties of FL by using a similar process as in (Li et al., 2020a). The datapoints $(\boldsymbol{x}_j, y_j)$ of device $i$ are generated based on $y_j = \arg\max(\boldsymbol{\theta}_i^* \boldsymbol{x}_j + \boldsymbol{b}_i^*)$ where $\boldsymbol{x}_j \in \mathbb{R}^{30\times1}$, $y_j \in \{1, 2, \ldots 5\}$, $\boldsymbol{\theta}_i^* \in \mathbb{R}^{5\times30}$, and $\boldsymbol{b}_i^* \in \mathbb{R}^{5\times1}$. $(\boldsymbol{\theta}_i^*, \boldsymbol{b}_i^*)$ tuple represents the optimal parameter set for device $i$ and each element of these tuples are randomly drawn from $\mathcal{N}(\mu_i, 1)$ where $\mu_i \sim \mathcal{N}(0, \gamma_1)$. The features of datapoints are modeled as $(\boldsymbol{x}_j \sim \mathcal{N}(\nu_i, \sigma))$ where $\sigma$ is a diagonal covariance matrix with elements $\sigma_{k,k} = k^{-1.2}$ and each element of $\nu_i$ is drawn from $\mathcal{N}(\beta_i, 1)$ where $\beta_i \sim \mathcal{N}(0, \gamma_2)$. The number of datapoints in device $i$ follows a lognormal distribution with variance $\gamma_3$. In this generation procees, $\gamma_1$, $\gamma_2$ and $\gamma_3$ regulate the relation of the optimal models for each device, the distribution of the features for each device and the amount of datapoints per device respectively.

We simulate different settings by allowing only one type of heterogeneity at a time and disabling the randomness from the other two. For instance, if we want to disable type 1 heterogeneity, we draw one single set of optimal parameters $(\boldsymbol{\theta}^*, \boldsymbol{b}^*) \sim \mathcal{N}(\boldsymbol{0}, \boldsymbol{1})$ and use it to generate datapoints for all devices. Similarly, $\nu_i$ is set to 0 to disable type 2 heterogeneity and $\gamma_3$ is set to 0 to disable type 3 heterogeneity. We consider four settings in total, including type 1, 2, and 3 heterogeneous as well as a homogeneous setting. The number of devices is set to 20 and the number of datapoints per device is on average 200 in the generation process.

**Models.** We test FedDyn, SCAFFOLD, FedAvg and FedProx using a multiclass logistic classification model with cross entropy loss. We keep batch size to be 10, weight decay to be $10^{-5}$.

We test learning rates in $[1, .1]$ and epochs in $[1, 10, 50]$ for all three algorithms. $\alpha$ parameter of FedDyn is chosen among $[.1, .01, .001]$; $K$ parameter of SCAFFOLD is searched in $[20, 200, 1000]$ which corresponds to the same amount of computation using above epoch list; and $\mu$ regularization hyperparameter of FedProx in $[0.01, .0001]$.

Table 6 reports the number models transmitted relative to one round of FedAvg to achieve the target training loss for best hyperparameter selection in various settings with $10\%$ device participation. As shown, FedDyn leads to communication savings in each of the settings in range $1.1\times$ to $7.6\times$.

## A.2    REAL DATA

**Datasets.** MNIST, EMNIST-L, CIFAR-10 and CIFAR-100 are used for image classification tasks and Shakespeare dataset is used for a next character prediction task. The image size is $(1 \times 28 \times 28)$ in MNIST and EMNIST; $(3 \times 32 \times 32)$ in CIFAR-10 and CIFAR-100 with overall 10 classes in MNIST and CIFAR-10; 62 classes in EMNIST; and 100 classes in CIFAR-100. We choose the first 10 letters from the letter section of EMNIST (named it as EMNIST-L) similar to (Li et al., 2020a) work. Features in Shakespeare dataset consists of 80 characters and labels are the following characters. Overall, there are 80 different labels for datapoints.

We use the usual train and test splits for MNIST, EMNIST-L, CIFAR-10 and CIFAR-100. The number of training and test samples of the benchmark datasets are summarized in Table 3.

To generate IID splits, we randomly divide training datapoints and assign them to devices. For non-IID splits, we utilize the Dirichlet distribution as in (Yurochkin et al., 2019). Firstly, a vector of size equal to the number of classes are drawn using Dirichlet distribution for each device. These vectors correspond to class priors per devices. Then one label is sampled based on these vectors for each device and an image is sampled without replacement based on the label. This process is repeated until all datapoints are assigned to devices. The procedure allows the label ratios of each device to follow a Dirichlet distribution. The hyperparameter of Dirichlet distribution corresponds to statistical heterogeneity level in the device datapoints. Overall, for a 100 device experiment, each device has 600, 480, 500 and 500 datapoints in MNIST, EMNIST-L, CIFAR-10 and CIFAR-100 respectively. For these datasets, three different federated settings are generated including an IID and two non-IID Dirichlet settings with .6 and .3 priors. Figure 3 shows the heterogeneity levels for MNIST dataset in

these different settings. The amount of most occurred class labels that consume $40\%$, $60\%$ and $80\%$ of device data are shown in the histogram plots. For example, every class label is equally represented in IID setting hence 4, 6 and 8 classes occupy $40\%$, $60\%$, and $80\%$ of the local datapoints for each device. If we consider non-IID settings, we see $80\%$ of local data belongs to mostly 4 or 5 different classes for Dirichlet .6; and 3 or 4 different classes for Dirichlet .3 settings.

To generate unbalanced data, we sample datapoint amounts from a lognormal distribution. Controlling the variance of lognormal distribution gives unbalanced data per devices. For instance, in CIFAR-10, balanced and unbalanced data settings have standard deviation of data amounts among devices as 0 and 0.3 respectively.

LEAF (Caldas et al., 2018) is used to generate the Shakespeare dataset used in this work. The LEAF framework allows to generate IID as well as non-IID federated settings. The non-IID dataset is the natural split of Shakespeare where each device corresponds to a role and the local dataset contains this role's sentences. The IID dataset is generated by combining the sentences from all roles and randomly dividing them into devices. In this work, we consider 100 devices and restrict number of datapoints per device to 2000.

**Models.** We use fully connected neural network architectures for MNIST and EMNIST-L. Both models take input images as a vector of 784 dimensions followed by 2 hidden layers and a final softmax layer. The number of neurons in the hidden layers are 200 and 100 for MNIST and EMNIST-L respectively. These models achieve $98.4\%$ and $95.0\%$ test accuracy in MNIST and EMNIST-L if trained on datapoints from all devices. The model considered for MNIST is the same model used in original FedAvg work (McMahan et al., 2017).

For CIFAR-10 and CIFAR-100, we use a CNN consisting of two convolutional layers with $64$ $5 \times 5$ filters, two $2 \times 2$ max pooling layers, two fully connected layers with 394 and 192 neurons, and finally a softmax layer. The models achieve $85.2\%$ and $55.3\%$ test accuracy in CIFAR-10 and CIFAR-100 respectively. Our CNN model is similar to the used for CIFAR-10 in the original FedAvg work (McMahan et al., 2017), except that we don't use Batch Normalization layers.

For the next character prediction task (Shakespeare), we use an LSTM. The model converts an 80 character long input sequence to a $80 \times 8$ sequence using an embedding. This sequence is fed to a two layer LSTM with hidden size of 100 units. The output of stacked LSTM is passed to a softmax layer. Overall, this architecture achieves a test accuracy of $50.8\%$ and $51.2\%$ in IID and non-IID settings, respectively, if trained on data from all devices. We report both IID and non-IID performance here because the datasets are randomly regenerated out of the whole Shakespeare writing hence train and test split is different for both cases. This Neural Network model is the same model used in the original FedProx study (Li et al., 2020a).

In passing, we note here that, we are not after state of the art model performances for these datasets, our aim is to compare the performances of these models in federated setting using FedDyn and other baselines.

**Hyperparameters.** We consider different hyperparameter configurations for different setups and datasets. For all the experiments, we fix batch size as 50 for MNIST, CIFAR-10, CIFAR-100 and EMNIST-L datasets and as 100 for Shakespeare dataset.

We note here that $\mu$, $\alpha$ and $K$ hyperparameters are used only in FedProx, FedDyn and SCAFFOLD respectively. $K$ is the equivalent of epoch for SCAFFOLD algorithm and we searched $K$ values to have the same amount of local computation as in other methods. For example, if each device has 500 datapoints, batch size is 50 and epoch is 10, local devices apply 100 SGD steps which is equivalent to $K$ being 100.

*MNIST.* As for the 100 devices, balanced data, full participation setup, hyperparameters are searched for all algorithms in all IID and Dirichlet settings for a fixed 100 communication rounds. The search space consists of learning rates in $[.1, .01]$, epochs in $[10, 20, 50]$, $K$s in $[120, 240, 600]$, $\mu$s in $[1, .01, .0001]$ and $\alpha$s in $[.001, .01, .03, .1]$. Weight decay of $10^{-4}$ is applied to prevent overfitting and no learning rate decay across communications rounds is used. The selected configuration for FedAvg is .1 learning rate and 20 epoch; for FedProx is .1 learning rate and .0001 $\mu$; for FedDyn is .1 learning rate, 50 epoch and .01 $\alpha$; and for SCAFFOLD is .1 learning rate and 600 $K$ for all IID and Dirichlet settings. These configurations are fixed and their performances are obtained for 500 communication rounds.

For the partial participation, 100 devices, balanced data setup, the selected configuration for FedAvg is .1 learning rate and 10 epoch; for FedProx is .1 learning rate and .0001 $\mu$; for FedDyn is .1 learning rate, 50 epoch and .01 $\alpha$; and for SCAFFOLD is .1 learning rate and 600 $K$ for all IID and Dirichlet settings except that $\alpha$ is chosen to be .03 for 10% IID setting. 0.998 learning rate decay per communication round is used and weight decay of $10^{-4}$ is applied to prevent overfitting for all methods.

For the centralized model, we choose learning rate as .1, epoch as 150 and learning rate is halved in every 50 epochs.

*EMNIST-L.* We used similar hyperparameters as in MNIST dataset. The configuration for FedAvg is .1 learning rate and 20 epoch; for FedProx is .1 learning rate and $10^{-4}$ $\mu$; for FedDyn is .1 learning rate, 50 epoch and 0.005 $\alpha$; and for SCAFFOLD is .1 learning rate and 500 $K$ for all IID and Dirichlet full participation settings.

The selected configuration for FedAvg is .1 learning rate and 10 epoch; for FedProx is .1 learning rate and .0001 $\mu$; for FedDyn is .1 learning rate, 50 epoch; and for SCAFFOLD is .1 learning rate and 500 $K$ for all IID and Dirichlet partial settings. $\alpha$ is chosen to be .003 for 10% and 1% IID; .005 for 10% Dirichlet .6 and 1% Dirichlet .3 ; .001 for 1% Dirichlet .6 and .01 for 10% Dirichlet .3 settings. 0.998 learning rate decay per communication round is used and weight decay of $10^{-4}$ is applied to prevent overfitting for all methods.

For the centralized model, we choose learning rate as .1, epoch as 150 and learning rate is halved in every 50 epochs.

*CIFAR-10.* The same hyperparameters are applied to all the CIFAR-10 experiments, including: 0.1 for learning rate, 5 for epochs, and $10^{-3}$ for weight decay. The learning rate decay is selected from the range of $[0.992, 0.998, 1.0]$. The $\alpha$ value is selected from the range of $[10^{-3}, 10^{-2}, 10^{-1}]$ for FedDyn. The $\mu$s value is selected from the range of $[10^{-2}, 10^{-3}, 10^{-4}]$.

For the centralized model, we choose learning rate as .1, epoch as 500 and learning rate decay as .992.

*CIFAR-100.* The same hyperparameters are applied to the CIFAR-100 experiments with 100 devices. including: 0.1 for learning rate, 5 for epochs, and $10^{-3}$ for weight decay. The learning rate decay is selected from the range of $[0.992, 0.998, 1.0]$. The $\alpha$ value is selected from the range of $[10^{-3}, 10^{-2}, 10^{-1}]$ for FedDyn. The $\mu$s value is selected from the range of $[10^{-2}, 10^{-3}, 10^{-4}]$.

As for 500 device, balanced data, 10% participation, IID setup, .1 learning rate, .0001 $\mu$, $10^{-3}$ weight decay applied. Epochs in $[2, 5]$ and corresponding $K$s in [4,10] searched. $\alpha$s in $[.1, .01, .001]$ are considered for FedDyn. Epoch of 2 is selected for FedDyn, FedAvg and FedProx, $K$ of 4 is selected for SCAFFOLD. .01 $\alpha$ value is selected for FedDyn. The same parameters are chosen for 500 device, balanced data, 10% participation, Dirichlet .3 setup.

As for 100 device, unbalanced data, 10% participation, IID and Dirichlet .3 settings, epoch of 2 is selected for FedDyn, FedAvg and FedProx, $K$ of 20 is selected for SCAFFOLD. .1 $\alpha$ value is applied for FedDyn. .0001 $\mu$ is used in FedProx.

For the centralized model, we choose learning rate as .1, epoch as 500 and learning rate decay as .992.

*Shakespeare.* As for 100 devices, balanced data, full participation setup, the hyperparameters are searched with all combinations of learning rate in $[1]$, epochs in $[1, 5]$, $K$s in $[20, 100]$, $\mu$s in $[.01, .0001]$ and $\alpha$s in $[.001, .009, .01, .015]$. Weight decay of $10^{-4}$ is applied to prevent overfitting and no learning rate decay across communications rounds is used. The selected configuration for FedAvg is 1 learning rate and 5 epoch; for FedProx is 1 learning rate, 5 epoch and .0001 $\mu$; for FedDyn is 1 learning rate, 5 epoch and .009 $\alpha$; and for SCAFFOLD is 1 learning rate and 100 $K$ in IID and non IID settings.

For the partial participation, 100 devices, balanced data setup, we choose 1 learning rate and 5 epoch for FedAvg; 1 learning rate, 5 epoch and .0001 $\mu$ for FedProx; 1 learning rate and 100 K for SCAFFOLD; and 1 learning rate and 5 epoch for FedDyn in all cases. $\alpha$ is .015 and .001 for 10% and 1% settings respectively. No learning rate decay is applied for 10% settings and a decay of .998 is applied for 1% settings. Weight decay of $10^{-4}$ is applied to prevent overfitting.

For the centralized model, we choose learning rate as 1, epoch as 150 and learning rate is halved in every 50 epochs.

Additionally, we performed gradient clipping to prevent overflow in weights for all methods. We found out that, this increases stability of algorithms.

**Convergence Plots.** We give convergence plots of experiments. The convergence plots of moderate and large number of devices in different device distributions are shown in Figure 4 and 5 for CIFAR-10 and CIFAR-100 datasets. Similarly, convergence curves of different participation levels and distributions are plotted in Figure 6, 7, 8, 9 and 10 for all datasets. Finally, Figure 12 and 13 show convergence plots for balanced data and unbalance data in different device distributions.

We emphasize that convergence curves show accuracy achieved with respect to rounds communicated. However, the metric we want to minimize, the amount of information transmitted, is not the same as number of communication rounds. For instance, SCAFFOLD transmits two models including state of devices per communication round. This difference is accounted in the tables.

We observed that averaging all device models gives more stable convergence curves hence we report the performance of the average model from all devices in each communication round. We note that we do not modify the algorithms, this part is only for reporting purposes.

Additional to experiments stated, we test our algorithm with a more complex model. We consider ResNet18 (He et al., 2016) structure on CIFAR-10 IID, 100 devices, balanced data, 10% participation setting. Batch normalization layers have inherent statistics which can be problematic in FL. Therefore, we use group normalization (Wu & He, 2018) instead of Batch normalization in ResNet18. The convergence curves are shown in Figure 11. FedDyn still outperforms the baseline methods in a higher capacity model setup.

### A.3 $\alpha$ SENSITIVITY ANALYSIS OF FEDDYN

$\alpha$ is an important parameter of FedDyn. Indeed, it is the only hyperparameter of the algorithm when devices have access to an optimization solver. In theory, $\alpha$ balances two problem dependent constants as shown in Theorem 2, Theorem 3 and Theorem 4. Consequently, optimal value of $\alpha$ depends on these constants. Since these constants are independent of $T$, the value of $\alpha$ does not asymptotically affect convergence rate.

To test sensitivity, we consider CIFAR-10, IID, 100 devices, 10% participation setting. Figure 1a shows convergence plots for different $\alpha$ configurations while keeping all other parameters constant in FedDyn. Figure 1b presents the best achieved test accuracy with respect to different $\alpha$ values. We see that best test performance is obtained when $\alpha = 10^{-1}$. We note that all configurations converge, but some of them converges to a better stationary points. This aligns with the theory because we guarantee convergence to a stationary point.

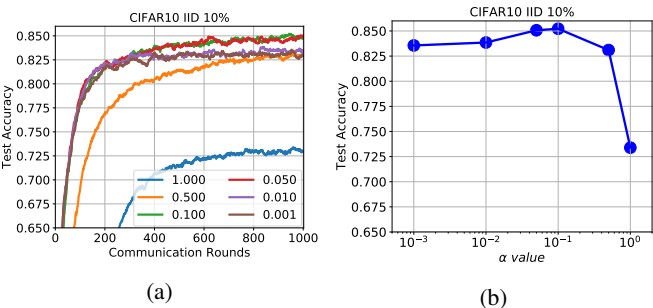

Figure 1: CIFAR-10 - $\alpha$ sensitivity analysis of FedDyn.

### A.4 COMPARISON TO A FULL PARTICIPATION METHOD

Recently, FedSplit (Pathak & Wainwright, 2020) is introduced to target non IID data distributions among devices. The work simplifies FL setting by considering full device participation. It charac-

terizes FedAvg convergence and shows that FedAvg should do only one step update per device in each round to achieve global minima if device losses are different. In such cases, FedAvg becomes decentralized SGD. After pointing out this inconsistency, FedSplit is given as a potential solution.

In this work, we aim to solve FL problem with four principle characteristic which are partial participation due to unreliable communication links, massive number of devices, heterogeneous device data and unbalanced data amounts per device. Partial participation is a critical property, because, it is inconceivable that we will not be in a situation where we have all devices participating in each round. However, FedSplit does not support partial participation.

Nevertheless, we adapt FedSplit to partial participation setting with the following changes. If a device is not active in the current round, its model $z_k^{t+1} = z_k^t$ and its intermediate state $z_k^{t+\frac{1}{2}} = z_k^{t-1+\frac{1}{2}}$ are frozen. For the server model, we have two options. First option is to keep the server model as average of all device models, $x^t = \frac{1}{m} \sum_{k \in m} z_k^t$, which is named as FedSplit All. Second option is to have the server model as the average of only current round's active devices $x^t = \frac{1}{|\mathcal{P}_t|} \sum_{k \in \mathcal{P}_t} z_k^t$, which is named as FedSplit Act. In passing, we do not claim that these modifications are optimal.

For empirical evaluation, we consider CIFAR-10, 100 devices, 100% and 10% participation settings. Figure 2a and 2b show comparison between FedSplit and FedDyn for 100% and 10% participation levels respectively. FedSplit All and FedSplit Act are the same in full participation setting hence shown as one method. We observe that FedDyn performs better than FedSplit in both cases. We see that FedSplit All where the server model averages all device models is significantly underperforming than FedSplit Act where the server only averages active devices. This is due to the fact that the server model is too slow to change when all devices are averaged because most of the devices are the same across consecutive rounds. We further note that it might not be easy to get convergence theory of FedSplit in the partial participation setting.

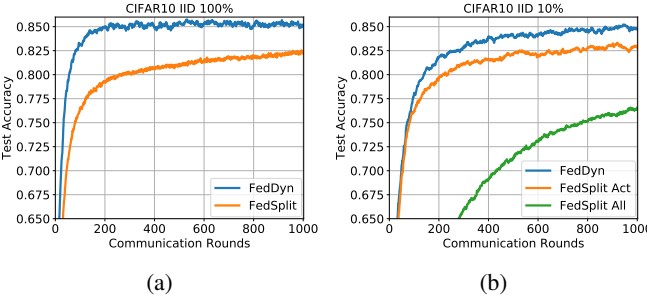

(a)  (b)

Figure 2: CIFAR-10 - FedSplit and FedDyn comparison in full and 10% participation settings.

## A.5 FIGURES OMITTED IN THE MAIN TEXT

Table 3: Datasets

| Dataset | Train Samples Amount | Test Samples Amount |
|---|---|---|
| CIFAR-10 | 50000 | 10000 |
| CIFAR-100 | 50000 | 10000 |
| MNIST | 60000 | 10000 |
| EMNIST-L | 48000 | 8000 |
| Shakespeare | 200000 | 40000 |

Table 4: Number of parameters transmitted relative to one round of FedAvg to reach target test accuracy for balanced data and unbalanced data in IID and Dirichlet .3 settings with $10\%$ participation. SCAFFOLD communicates the current model and its associated gradient per round, while others communicate only the current model. As such number of rounds for SCAFFOLD is one half of those reported.

| Local Data | Dataset | Accuracy | FedDyn | SCAFFOLD | FedAvg | FedProx |
|---|---|---|---|---|---|---|
| **Balanced** | | | | **IID** | | |
| | CIFAR-10 | 84.5 | 637 | 1852(2.9×) | 1000+(>1.6×) | 1000+(>1.6×) |
| | | 82.3 | 240 | 512(2.1×) | 994(4.1×) | 825(3.4×) |
| | CIFAR-100 | 51.0 | 522 | 1854(3.6×) | 1000+(>1.9×) | 1000+(>1.9×) |
| | | 40.9 | 159 | 286(1.8×) | 822(5.2×) | 873(5.5×) |
| | | | | **Dirichlet (.3)** | | |
| | CIFAR-10 | 82.5 | 444 | 1880(4.2×) | 1000+(>2.3×) | 1000+(>2.3×) |
| | | 80.7 | 232 | 594(2.6×) | 863(3.7×) | 930(4.0×) |
| | CIFAR-100 | 51.0 | 561 | 1884(3.4×) | 1000+(>1.8×) | 1000+(>1.8×) |
| | | 42.3 | 170 | 330(1.9×) | 959(5.6×) | 882(5.2×) |
| **Unbalanced** | | | | **IID** | | |
| | CIFAR-10 | 84.0 | 335 | 1152(3.4×) | 1000+(>3.0×) | 1000+(>3.0×) |
| | | 82.3 | 213 | 548(2.6×) | 834(3.9×) | 834(3.9×) |
| | CIFAR-100 | 53.0 | 386 | 1656(4.3×) | 1000+(>2.6×) | 1000+(>2.6×) |
| | | 48.2 | 209 | 800(3.8×) | 968(4.6×) | 945(4.5×) |
| | | | | **Dirichlet (.3)** | | |
| | CIFAR-10 | 82.5 | 524 | 1998(3.8×) | 1000+(>1.9×) | 1000+(>1.9×) |
| | | 80.1 | 274 | 652(2.4×) | 893(3.3×) | 1000+(>3.6×) |
| | CIFAR-100 | 52.0 | 503 | 1928(3.8×) | 1000+(>2.0×) | 1000+(>2.0×) |
| | | 47.3 | 234 | 942(4.0×) | 871(3.7×) | 1000+(>4.3×) |

Table 5: Number of parameters transmitted relative to one round of FedAvg to reach target test accuracy for 1% participation regime in the IID, non-IID settings. SCAFFOLD communicates the current model and its associated gradient per round, while others communicate only the current model. As such number of rounds for SCAFFOLD is one half of those reported.

| Participation | Dataset | Accuracy | FedDyn | SCAFFOLD | FedAvg | FedProx |
|---|---|---|---|---|---|---|
| | | **IID** | | | | |
| | CIFAR-10 | 82.6 | 660 | 1544(2.3×) | 892(1.4×) | 1000+(>1.5×) |
| | | 81.6 | 543 | 1150(2.1×) | 603(1.1×) | 707(1.3×) |
| | CIFAR-100 | 39.8 | 409 | 1982(4.8×) | 428(1.0×) | 512(1.3×) |
| | | 38.8 | 396 | 1862(4.7×) | 392(1.0×) | 454(1.1×) |
| | MNIST | 98.3 | 529 | 956(1.8×) | 644(1.2×) | 451(0.9×) |
| | | 97.3 | 145 | 290(2.0×) | 151(1.0×) | 143(1.0×) |
| | EMNIST-L | 94.9 | 483 | 1136(2.4×) | 826(1.7×) | 1000+(>2.1×) |
| | | 93.9 | 210 | 554(2.6×) | 216(1.0×) | 238(1.1×) |
| | Shakespeare | 43.0 | 170 | 460(2.7×) | 188(1.1×) | 151(0.9×) |
| | | 42.0 | 148 | 342(2.3×) | 149(1.0×) | 142(1.0×) |
| | | **Dirichlet (.6)** | | | | |
| | CIFAR-10 | 81.0 | 561 | 1510(2.7×) | 977(1.7×) | 841(1.5×) |
| | | 80.0 | 436 | 1100(2.5×) | 673(1.5×) | 623(1.4×) |
| | CIFAR-100 | 36.6 | 355 | 1996(5.6×) | 341(1.0×) | 352(1.0×) |
| | | 35.6 | 342 | 1876(5.5×) | 317(0.9×) | 342(1.0×) |
| 1% | MNIST | 98.2 | 486 | 1502(3.1×) | 863(1.8×) | 754(1.6×) |
| | | 97.2 | 180 | 332(1.8×) | 199(1.1×) | 166(0.9×) |
| | EMNIST-L | 94.8 | 405 | 1230(3.0×) | 504(1.2×) | 1000+(>2.5×) |
| | | 93.8 | 195 | 576(3.0×) | 256(1.3×) | 294(1.5×) |
| | | **Dirichlet (.3)** | | | | |
| | CIFAR-10 | 79.0 | 590 | 1580(2.7×) | 955(1.6×) | 738(1.3×) |
| | | 78.0 | 452 | 1272(2.8×) | 653(1.4×) | 497(1.1×) |
| | CIFAR-100 | 36.1 | 343 | 1990(5.8×) | 317(0.9×) | 342(1.0×) |
| | | 35.1 | 321 | 1866(5.8×) | 294(0.9×) | 314(1.0×) |
| | MNIST | 98.2 | 521 | 954(1.8×) | 951(1.8×) | 974(1.9×) |
| | | 97.2 | 157 | 318(2.0×) | 169(1.1×) | 177(1.1×) |
| | EMNIST-L | 94.4 | 442 | 1860(4.2×) | 481(1.1×) | 1000+(>2.3×) |
| | | 93.4 | 241 | 694(2.9×) | 286(1.2×) | 279(1.2×) |
| | | **Non-IID** | | | | |
| | Shakespeare | 43.8 | 158 | 388(2.5×) | 159(1.0×) | 153(1.0×) |
| | | 42.8 | 143 | 318(2.2×) | 146(1.0×) | 145(1.0×) |

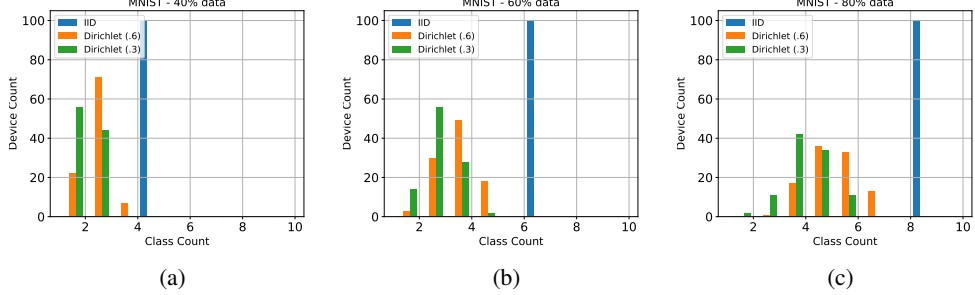

(a)          (b)          (c)

Figure 3: MNIST- Histogram of device counts whose 40% (3a), 60% (3b), and 80% (3c) datapoints belong to $k$ classes.

Table 6: Number of parameters transmitted relative to one round of FedAvg to reach target test accuracy for convex synthetic problem in different types of heterogeneity settings. SCAFFOLD communicates the current model and its associated gradient per round, while others communicate only the current model. As such number of rounds for SCAFFOLD is one half of those reported.

| Loss | FedDyn | SCAFFOLD | FedAvg | FedProx |
|------|--------|----------|--------|---------|
| **Homogeneous** | | | | |
| 0.0603 | 32 | 70(2.2×) | 136(4.2×) | 49(1.5×) |
| **Type 1 Heterogeneous** | | | | |
| 1.5717 | 17 | 88(5.2×) | 20(1.2×) | 18(1.1×) |
| **Type 2 Heterogeneous** | | | | |
| 0.1205 | 150 | 164(1.1×) | 274(1.8×) | 275(1.8×) |
| **Type 3 Heterogeneous** | | | | |
| 0.0854 | 34 | 260(7.6×) | 79(2.3×) | 106(3.1×) |

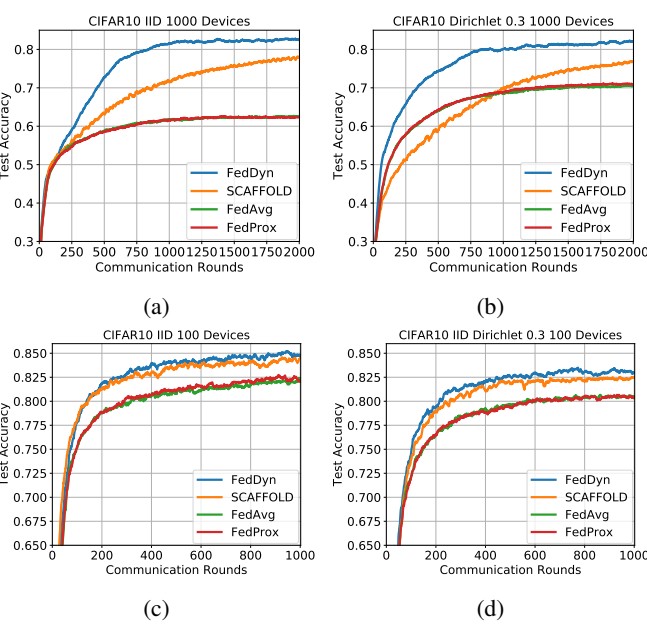

Figure 4: CIFAR-10- Convergence curves for different $100$ and $1000$ devices in the IID and Dirichlet (.3) settings with $10\%$ participation level and balanced data.

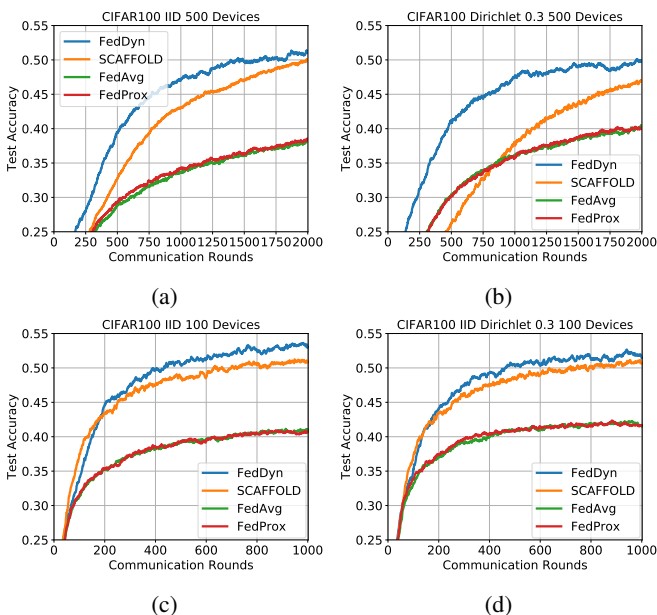

Figure 5: CIFAR-100- Convergence curves for different 100 and 500 devices in the IID and Dirichlet (.3) settings with 10% participation level and balanced data.

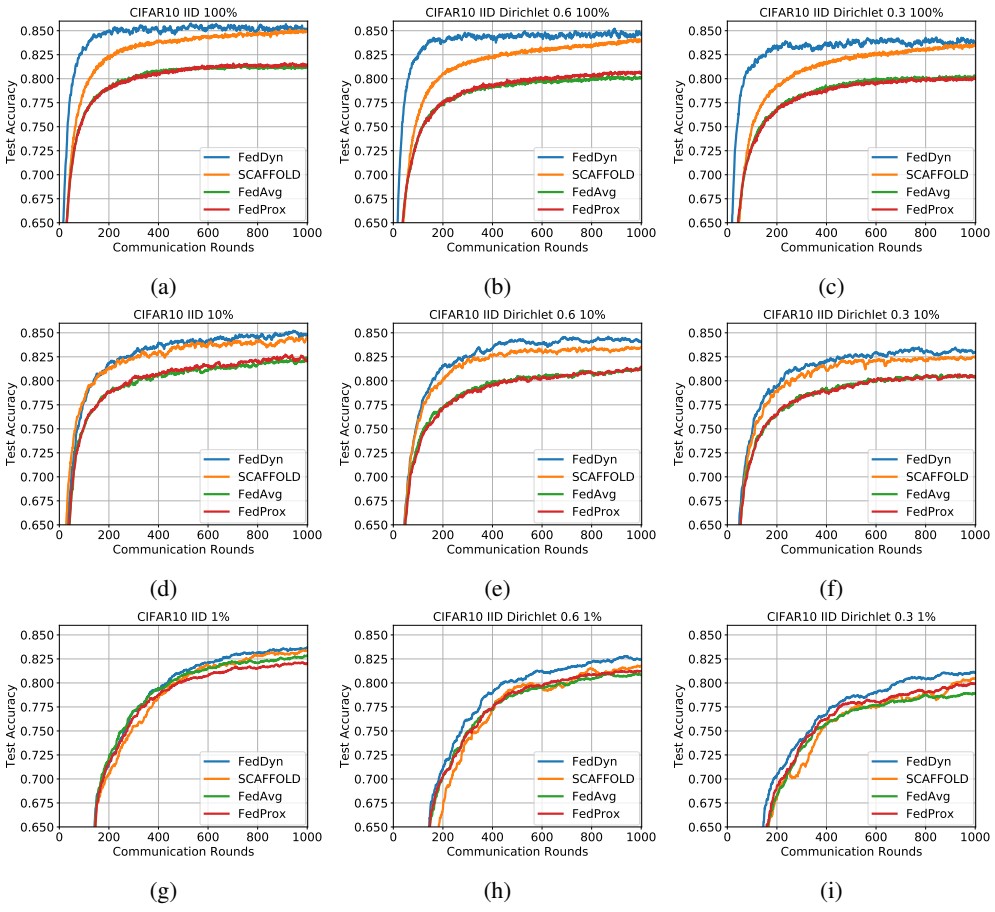

Figure 6: CIFAR-10- Convergence curves for participation fractions ranging from 100% to 10% to 1% in the IID, Dirichlet (.6) and Dirichlet (.3) settings with 100 devices and balanced data.

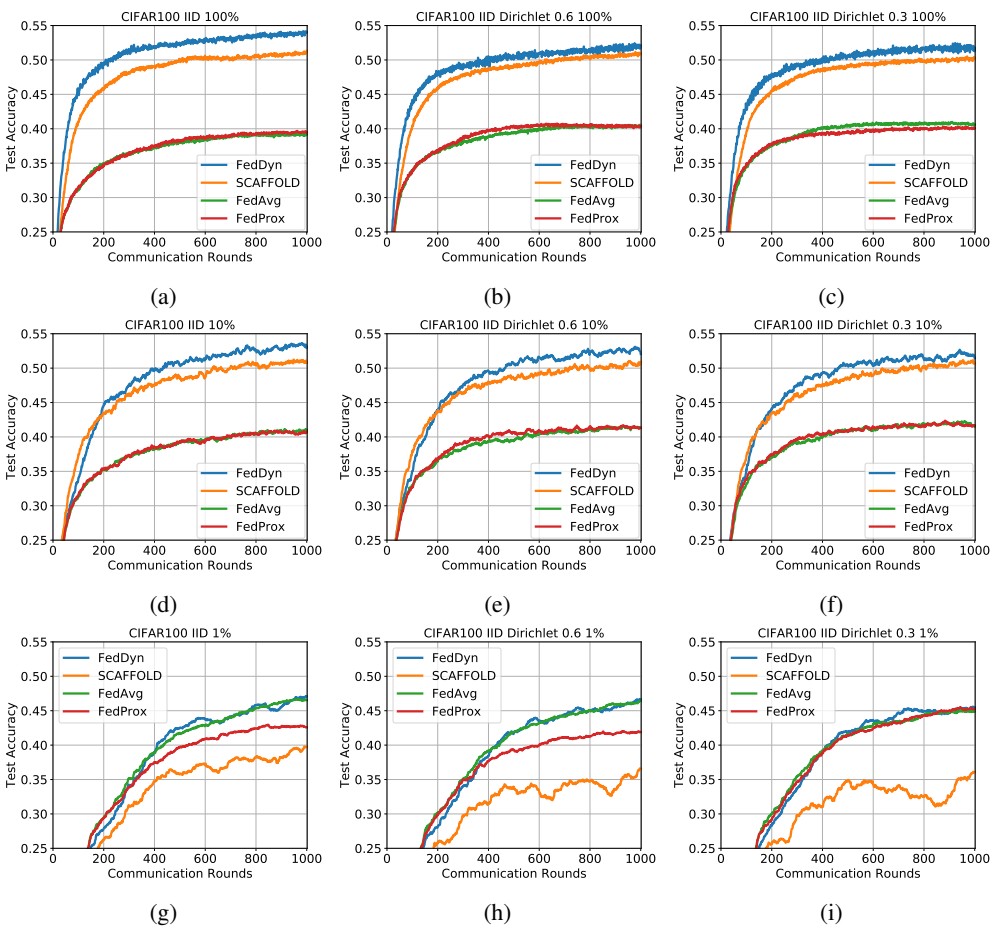

Figure 7: CIFAR-100- Convergence curves for participation fractions ranging from 100% to 10% to 1% in the IID, Dirichlet (.6) and Dirichlet (.3) settings with 100 devices and balanced data.

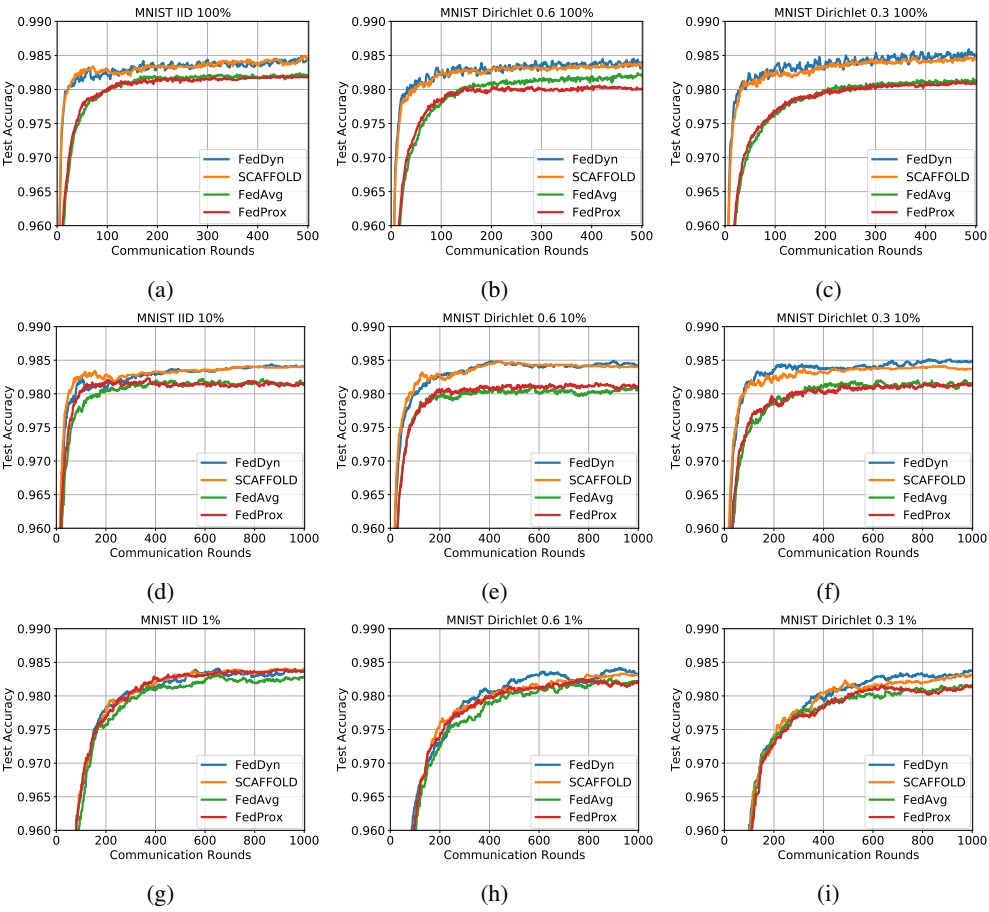

(a)           (b)           (c)

(d)           (e)           (f)

(g)           (h)           (i)

Figure 8: MNIST- Convergence curves for participation fractions ranging from 100% to 10% to 1% in the IID, Dirichlet (.6) and Dirichlet (.3) settings with 100 devices and balanced data.

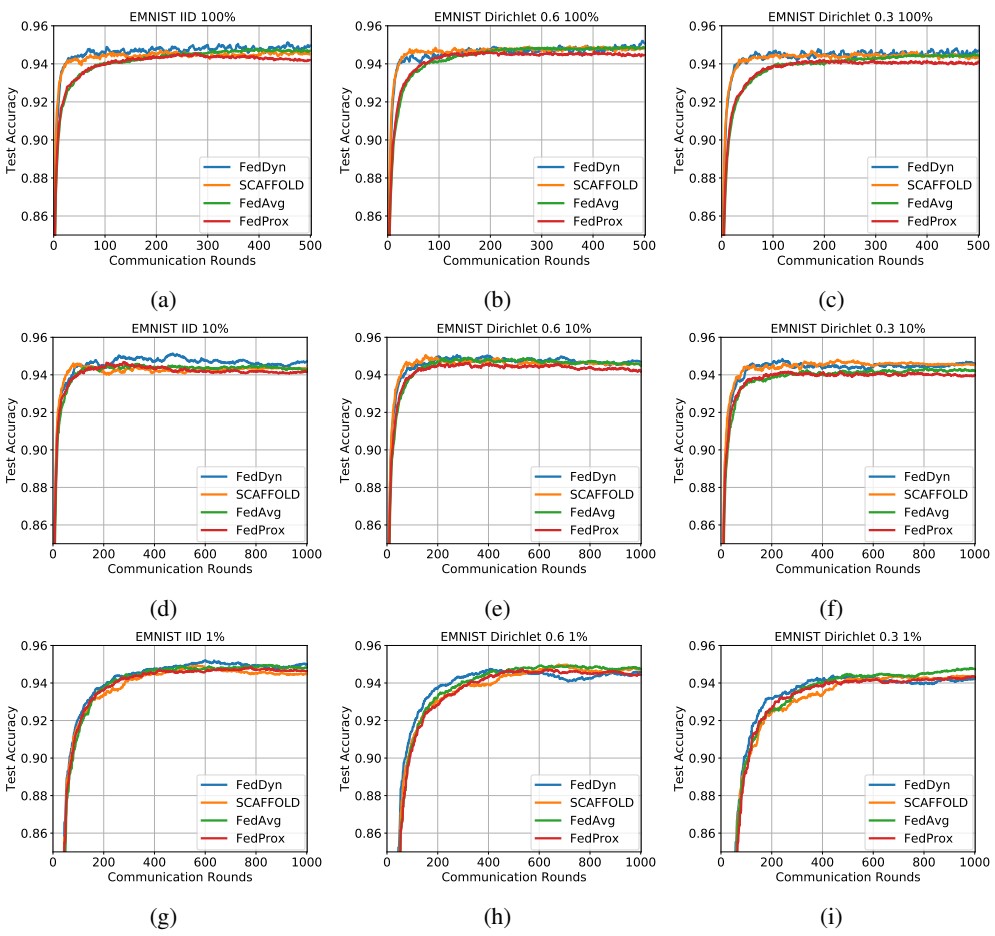

Figure 9: EMNIST-L- Convergence curves for participation fractions ranging from 100% to 10% to 1% in the IID, Dirichlet (.6) and Dirichlet (.3) settings with 100 devices and balanced data.

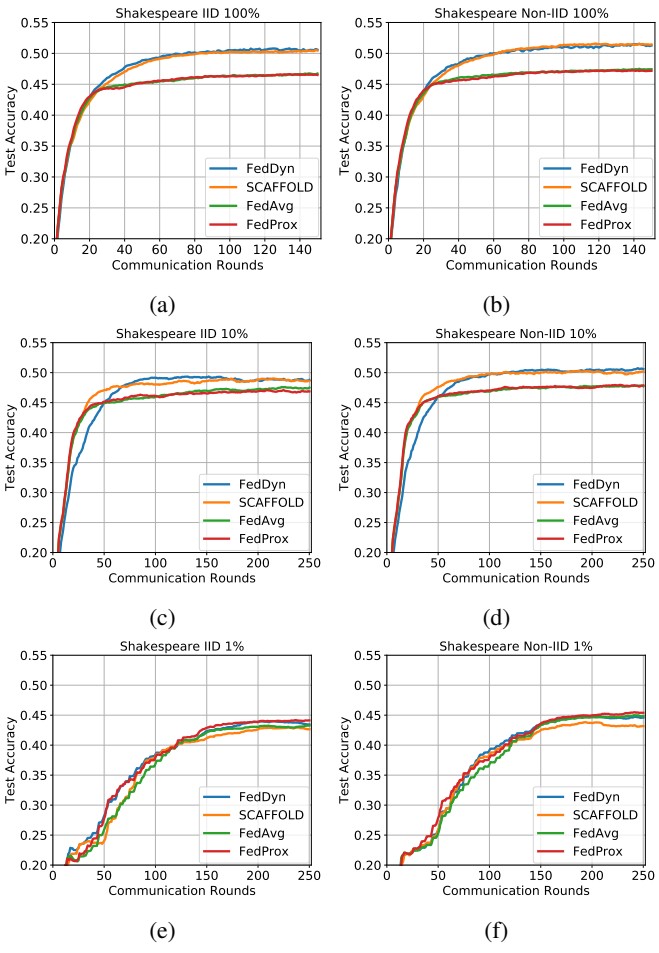

(a)  (b)

(c)  (d)

(e)  (f)

Figure 10: Shakespeare- Convergence curves for participation fractions ranging from 100% to 10% to 1% in the IID, and non-IID settings with 100 devices and balanced data.

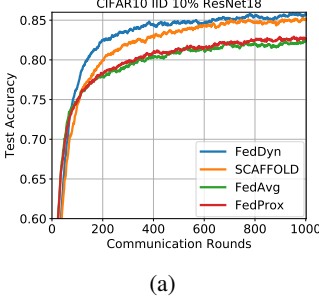

(a)

Figure 11: Convergence curves for ResNet18 with 1000 devices and balanced data.

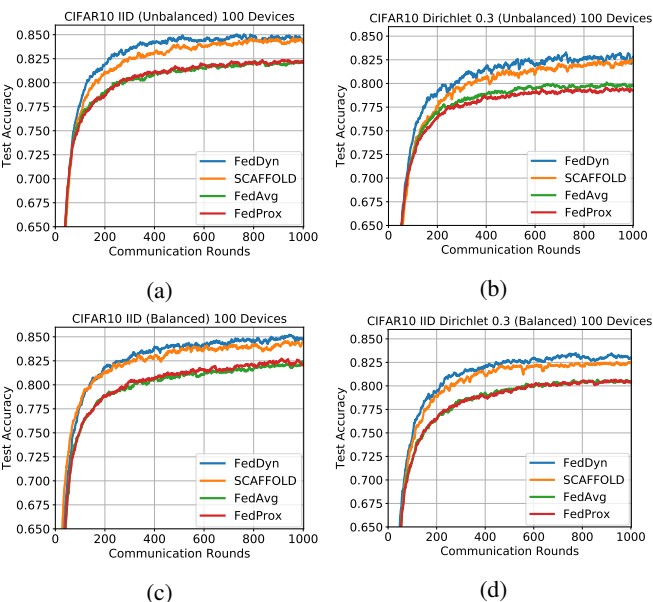

Figure 12: CIFAR-10- Convergence curves for balanced and unbalanced data distributions with 10% participation level as well as 100 devices in the IID and Dirichlet (.3) settings.

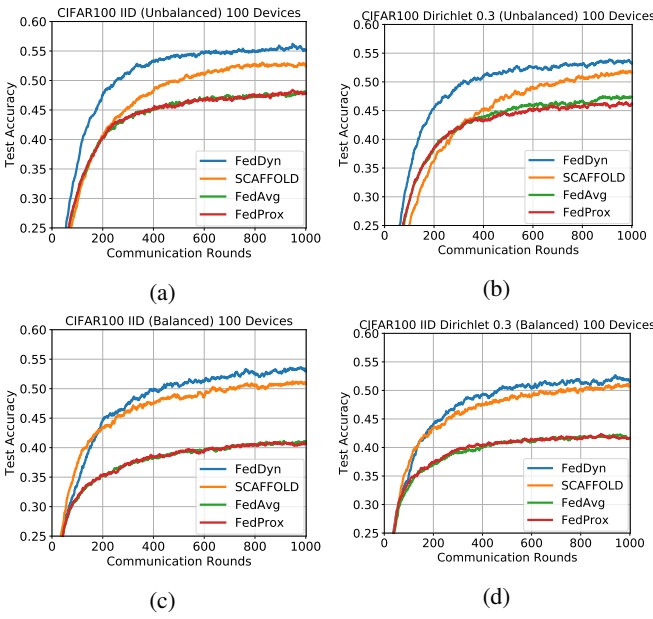

Figure 13: CIFAR-100- Convergence curves for balanced and unbalanced data distributions with 10% participation level as well as 100 devices in the IID and Dirichlet (.3) settings.

## B    PROOF

### B.1    CONVEX ANALYSIS

**Definition 1.** $L_k$ *is L smooth if*

$$\|\nabla L_k(\boldsymbol{x}) - \nabla L_k(\boldsymbol{y})\| \leq L\|\boldsymbol{x} - \boldsymbol{y}\| \quad \forall \boldsymbol{x}, \boldsymbol{y}$$

Smoothness implies the following quadratic bound,

$$L_k(\boldsymbol{y}) \leq L_k(\boldsymbol{x}) + \langle \nabla L_k(\boldsymbol{x}), \boldsymbol{y} - \boldsymbol{x}\rangle + \frac{L}{2}\|\boldsymbol{y} - \boldsymbol{x}\|^2 \quad \forall \boldsymbol{x}, \boldsymbol{y} \tag{4}$$

If $\{L_k\}_{k=1}^m$s are convex and $L$ smooth we have

$$\frac{1}{2Lm} \sum_{k \in [m]} \|\nabla L_k(\boldsymbol{x}) - \nabla L_k(\boldsymbol{x}_*)\|^2 \leq \ell(\boldsymbol{x}) - \ell(\boldsymbol{x}_*) \quad \forall \boldsymbol{x} \tag{5}$$

$$-\langle \nabla L_k(\boldsymbol{x}), \boldsymbol{z} - \boldsymbol{y}\rangle \leq -L_k(\boldsymbol{z}) + L_k(\boldsymbol{y}) + \frac{L}{2}\|\boldsymbol{z} - \boldsymbol{x}\|^2 \quad \forall \boldsymbol{x}, \boldsymbol{y}, \boldsymbol{z} \tag{6}$$

where $\ell(\boldsymbol{x}) = \frac{1}{m}\sum_{k=1}^m L_k(\boldsymbol{x})$ and $\nabla \ell(\boldsymbol{x}_*) = \boldsymbol{0}$.

We state convergence as,

**Theorem 2.** *For convex and L smooth $\{L_k\}_{k=1}^m$ functions and $\alpha \geq 25L$, Algorithm 1 satisfies*

$$E\left[\ell\left(\frac{1}{T}\sum_{t=0}^{T-1}\boldsymbol{\gamma}^t\right) - \ell(\boldsymbol{\theta}_*)\right] \leq \frac{1}{T}\left(10\alpha\left\|\boldsymbol{\theta}^0 - \boldsymbol{\theta}_*\right\|^2 + 100\frac{m}{P}\frac{1}{\alpha}\left(\frac{1}{m}\sum_{k\in[m]}\|\nabla L_k(\boldsymbol{\theta}_*)\|^2\right)\right) = O\left(\frac{1}{T}\right)$$

*where $\boldsymbol{\gamma}^t = \frac{1}{P}\sum_{k\in\mathcal{P}_t}\boldsymbol{\theta}_k^t$, $\boldsymbol{\theta}_* = \arg\min_{\boldsymbol{\theta}}\ell(\boldsymbol{\theta})$.*

If $\alpha = 30L\sqrt{\frac{m}{P}}$, we get the statement in Theorem 1. Throughout the proof, we utilize similar techniques as in SCAFFOLD (Karimireddy et al., 2019) convergence. We define a set of variables which are useful in the analysis. Algorithm 1 freezes $\boldsymbol{\theta}_k$ and its gradients if the device is not active. Let's define virtual $\{\tilde{\boldsymbol{\theta}}_k^t\}$ variables as

$$\tilde{\boldsymbol{\theta}}_k^t = \arg\min_{\boldsymbol{\theta}} L_k(\boldsymbol{\theta}) - \langle \nabla L_k(\boldsymbol{\theta}_k^{t-1}), \boldsymbol{\theta}\rangle + \frac{\alpha}{2}\|\boldsymbol{\theta} - \boldsymbol{\theta}^{t-1}\|^2 \quad \forall k \in [m], t > 0 \tag{7}$$

We see that $\tilde{\boldsymbol{\theta}}_k^t = \boldsymbol{\theta}_k^t$ if $k \in \mathcal{P}_t$ and $\tilde{\boldsymbol{\theta}}_k^t$ doesn't depend on $\mathcal{P}_t$. First order condition in Eq. 7 and in device optimization give

$$\tilde{\boldsymbol{\theta}}_k^t - \boldsymbol{\theta}^{t-1} = \frac{1}{\alpha}(\nabla L_k(\boldsymbol{\theta}_k^{t-1}) - \nabla L_k(\tilde{\boldsymbol{\theta}}_k^t)) \; \forall k \in [m]; \;\; \boldsymbol{\theta}_k^t - \boldsymbol{\theta}^{t-1} = \frac{1}{\alpha}(\nabla L_k(\boldsymbol{\theta}_k^{t-1}) - \nabla L_k(\boldsymbol{\theta}_k^t)) \; \forall k \in \mathcal{P}_t \tag{8}$$

$\boldsymbol{\theta}^t$ consists of active device average and gradient parts. Let's express active device average and its relation with the server model as,

$$\boldsymbol{\gamma}^t = \frac{1}{P}\sum_{k\in\mathcal{P}_t}\boldsymbol{\theta}_k^t; \;\; \boldsymbol{\gamma}^t = \boldsymbol{\theta}^t + \frac{1}{\alpha}\boldsymbol{h}^t \tag{9}$$

Due to linear update of $\nabla L_k$, $\boldsymbol{h}$ state in the server becomes as $\boldsymbol{h}^t = \frac{1}{m}\sum_{k\in[m]}\nabla L_k(\boldsymbol{\theta}_k^t)$.

Let's define some quantities that we would like to control.

$$C_t = \frac{1}{m}\sum_{k\in[m]} E\|\nabla L_k(\boldsymbol{\theta}_k^t) - \nabla L_k(\boldsymbol{\theta}_*)\|^2, \;\; \epsilon_t = \frac{1}{m}\sum_{k\in[m]} E\|\tilde{\boldsymbol{\theta}}_k^t - \boldsymbol{\gamma}^{t-1}\|^2$$

$C_t$ tracks how well local gradients of device models approximate the gradient of optimal model. If models converge to $\boldsymbol{\theta}_*$, $C_t$ will be 0. $\epsilon_t$ keeps track of how much local models change compared to average of device models from previous round. Again, upon convergence $\epsilon_t$ will be 0.

After these definitions, Theorem 2 can be seen as a direct consequence of the following Lemma,

**Lemma 1.** *For convex and L smooth $\{L_k\}_{k=1}^m$ functions, if $\alpha \geq 25L$, Algorithm 1 satisfies*

$$E\|\gamma^t - \boldsymbol{\theta}_*\|^2 + \kappa C_t \leq E\|\gamma^{t-1} - \boldsymbol{\theta}_*\|^2 + \kappa C_{t-1} - \kappa_0 E\left[\ell(\gamma^{t-1}) - \ell(\boldsymbol{\theta}_*)\right]$$

*where $\kappa = 8\frac{m}{P}\frac{1}{\alpha}\frac{L+\alpha}{\alpha^2 - 20L^2}, \kappa_0 = 2\frac{1}{\alpha}\frac{\alpha^2 - 20\alpha L - 40L^2}{\alpha^2 - 20L^2}$*

Lemma 1 can be telescoped in the following way,

$$\kappa_0 E\left[\ell(\gamma^{t-1}) - \ell(\boldsymbol{\theta}_*)\right] \leq \left(E\|\gamma^{t-1} - \boldsymbol{\theta}_*\|^2 + \kappa C_{t-1}\right) - \left(E\|\gamma^t - \boldsymbol{\theta}_*\|^2 + \kappa C_t\right)$$

$$\kappa_0 \sum_{t=1}^T E\left[\ell(\gamma^{t-1}) - \ell(\boldsymbol{\theta}_*)\right] \leq \left(E\|\gamma^0 - \boldsymbol{\theta}_*\|^2 + \kappa C_0\right) - \left(E\|\gamma^T - \boldsymbol{\theta}_*\|^2 + \kappa C_T\right)$$

If $\alpha \geq 25L$, $\kappa_0$ and $\kappa$ become positive. By definition, we also have $C_t$ sequences as positive. Eliminating negative terms on RHS gives,

$$\kappa_0 \sum_{t=1}^T E\left[\ell(\gamma^{t-1}) - \ell(\boldsymbol{\theta}_*)\right] \leq E\|\gamma^0 - \boldsymbol{\theta}_*\|^2 + \kappa C_0$$

Applying Jensen on LHS gives,

$$E\left[\ell\left(\frac{1}{T}\sum_{t=1}^T \gamma^{t-1}\right) - \ell(\boldsymbol{\theta}_*)\right] \leq \frac{1}{T}\frac{1}{\kappa_0}\left(\|\gamma^0 - \boldsymbol{\theta}_*\|^2 + \kappa C_0\right) = O\left(\frac{1}{T}\right)$$

which proves the statement in Theorem 2.

Similar to fundamental gradient descent analysis, $\|\gamma^t - \boldsymbol{\theta}_*\|^2$ is expressed as $\|\gamma^t - \gamma^{t-1} + \gamma^{t-1} - \boldsymbol{\theta}_*\|^2$ and expanded in the proof of Lemma 1. The resulting expression has $(\gamma^t - \gamma^{t-1})$ and $\|\gamma^t - \gamma^{t-1}\|^2$ terms. To tackle these extra terms, we state the following Lemmas and prove long ones at the end.

**Lemma 2.** *Algorithm 1 satisfies*

$$E\left[\gamma^t - \gamma^{t-1}\right] = \frac{1}{\alpha m}\sum_{k \in [m]} E\left[-\nabla L_k(\tilde{\boldsymbol{\theta}}_k^t)\right]$$

*Proof.*

$$E\left[\gamma^t - \gamma^{t-1}\right] = E\left[\left(\frac{1}{P}\sum_{k \in \mathcal{P}_t}\boldsymbol{\theta}_k^t\right) - \boldsymbol{\theta}^{t-1} - \frac{1}{\alpha}\boldsymbol{h}^{t-1}\right] = E\left[\frac{1}{P}\sum_{k \in \mathcal{P}_t}\left(\boldsymbol{\theta}_k^t - \boldsymbol{\theta}^{t-1} - \frac{1}{\alpha}\boldsymbol{h}^{t-1}\right)\right]$$

$$= E\left[\frac{1}{\alpha P}\sum_{k \in \mathcal{P}_t}\left(\nabla L_k(\boldsymbol{\theta}_k^{t-1}) - \nabla L_k(\boldsymbol{\theta}_k^t) - \boldsymbol{h}^{t-1}\right)\right]$$

$$= E\left[\frac{1}{\alpha P}\sum_{k \in \mathcal{P}_t}\left(\nabla L_k(\boldsymbol{\theta}_k^{t-1}) - \nabla L_k(\tilde{\boldsymbol{\theta}}_k^t) - \boldsymbol{h}^{t-1}\right)\right]$$

$$= E\left[\frac{1}{\alpha m}\sum_{k \in [m]}\left(\nabla L_k(\boldsymbol{\theta}_k^{t-1}) - \nabla L_k(\tilde{\boldsymbol{\theta}}_k^t) - \boldsymbol{h}^{t-1}\right)\right] = \frac{1}{\alpha m}\sum_{k \in [m]} E\left[-\nabla L_k(\tilde{\boldsymbol{\theta}}_k^t)\right]$$

where first equation is from definition in Eq. 9. The following equations come from Eq. 8 and $\tilde{\boldsymbol{\theta}}_k^t = \boldsymbol{\theta}_k^t$ if $k \in \mathcal{P}_t$ respectively. Fifth equation is due to taking expectation while conditioning on randomness before time $t$. If conditioned on randomness prior to $t$, every variable except $\mathcal{P}_t$ is revealed and each device is selected with probability $\frac{P}{m}$. Last one is due to definition of $\boldsymbol{h}^t = \frac{1}{m}\sum_{k \in [m]} \nabla L_k(\boldsymbol{\theta}_k^t)$. $\square$

Similarly, $\|\gamma^t - \gamma^{t-1}\|^2$ is bounded with the following,

**Lemma 3.** *Algorithm 1 satisfies*

$$E\|\gamma^t - \gamma^{t-1}\|^2 \leq \epsilon_t$$

*Proof.*

$$E\|\boldsymbol{\gamma}^t - \boldsymbol{\gamma}^{t-1}\|^2 = E\left\|\frac{1}{P}\sum_{k\in\mathcal{P}_t}(\boldsymbol{\theta}_k^t - \boldsymbol{\gamma}^{t-1})\right\|^2 \leq \frac{1}{P}E\left[\sum_{k\in\mathcal{P}_t}\|\boldsymbol{\theta}_k^t - \boldsymbol{\gamma}^{t-1}\|^2\right] = \frac{1}{P}E\left[\sum_{k\in\mathcal{P}_t}\left\|\tilde{\boldsymbol{\theta}}_k^t - \boldsymbol{\gamma}^{t-1}\right\|^2\right]$$

$$= \frac{1}{P}\frac{P}{m}\sum_{k\in[m]}E\left\|\tilde{\boldsymbol{\theta}}_k^t - \boldsymbol{\gamma}^{t-1}\right\|^2 = \epsilon_t$$

where first equality comes from Eq. 9. The following inequality applies Jensen. Remaining relations are due to $\tilde{\boldsymbol{\theta}}_k^t = \boldsymbol{\theta}_k^t$ if $k \in \mathcal{P}_t$, taking expectation by conditioning on randomness before time $t$ and definition of $\epsilon_t$.□

We need to further bound excess $\epsilon_t$ term arising in Lemma 3. We introduce two more Lemmas to handle this term.

**Lemma 4.** *For convex and L smooth $\{L_k\}_{k=1}^m$ functions, Algorithm 1 satisfies*

$$\left(1 - 4L^2\frac{1}{\alpha^2}\right)\epsilon_t \leq 8\frac{1}{\alpha^2}C_{t-1} + 8L\frac{1}{\alpha^2}E\left[\ell(\boldsymbol{\gamma}^{t-1}) - \ell(\boldsymbol{\theta}_*)\right]$$

**Lemma 5.** *For convex and L smooth $\{L_k\}_{k=1}^m$ functions, Algorithm 1 satisfies*

$$C_t \leq \left(1 - \frac{P}{m}\right)C_{t-1} + 2L^2\frac{P}{m}\epsilon_t + 4L\frac{P}{m}E\left[\ell(\boldsymbol{\gamma}^{t-1}) - \ell(\boldsymbol{\theta}_*)\right]$$

$E\left[\ell(\boldsymbol{\gamma}^{t-1}) - \ell(\boldsymbol{\theta}_*)\right]$ terms constitute LHS of the telescopic sum. Let's express $\|\boldsymbol{\gamma}^t - \boldsymbol{\theta}_*\|^2$ term as,

$$E\|\boldsymbol{\gamma}^t - \boldsymbol{\theta}_*\|^2 = E\|\boldsymbol{\gamma}^{t-1} - \boldsymbol{\theta}_* + \boldsymbol{\gamma}^t - \boldsymbol{\gamma}^{t-1}\|^2$$

$$= E\|\boldsymbol{\gamma}^{t-1} - \boldsymbol{\theta}_*\|^2 + 2E\left[\left\langle\boldsymbol{\gamma}^{t-1} - \boldsymbol{\theta}_*, \boldsymbol{\gamma}^t - \boldsymbol{\gamma}^{t-1}\right\rangle\right] + E\|\boldsymbol{\gamma}^t - \boldsymbol{\gamma}^{t-1}\|^2$$

$$= E\|\boldsymbol{\gamma}^{t-1} - \boldsymbol{\theta}_*\|^2 + \frac{2}{\alpha m}\sum_{k\in[m]}E\left[\left\langle\boldsymbol{\gamma}^{t-1} - \boldsymbol{\theta}_*, -\nabla L_k(\tilde{\boldsymbol{\theta}}_k^t)\right\rangle\right] + E\|\boldsymbol{\gamma}^t - \boldsymbol{\gamma}^{t-1}\|^2$$

$$\leq E\|\boldsymbol{\gamma}^{t-1} - \boldsymbol{\theta}_*\|^2 + \frac{2}{\alpha m}\sum_{k\in[m]}E\left[L_k(\boldsymbol{\theta}_*) - L_k(\boldsymbol{\gamma}^{t-1}) + \frac{L}{2}\|\tilde{\boldsymbol{\theta}}_k^t - \boldsymbol{\gamma}^{t-1}\|^2\right]$$

$$+ E\|\boldsymbol{\gamma}^t - \boldsymbol{\gamma}^{t-1}\|^2$$

$$= E\|\boldsymbol{\gamma}^{t-1} - \boldsymbol{\theta}_*\|^2 - \frac{2}{\alpha}E\left[\ell(\boldsymbol{\gamma}^{t-1}) - \ell(\boldsymbol{\theta}_*)\right] + \frac{L}{\alpha}\epsilon_t + E\|\boldsymbol{\gamma}^t - \boldsymbol{\gamma}^{t-1}\|^2 \qquad (10)$$

where we first expand the square term and use Lemma 2. Following inequality is due to Inq. 6.

Let's scale Lemma 4 and 5 with $\alpha\frac{L+\alpha}{\alpha^2-20L^2}$ and $8\frac{m}{P}\frac{1}{\alpha}\frac{L+\alpha}{\alpha^2-20L^2}$ respectively. We note that the coefficients are positive due to the condition on $\alpha$. Summing Inq. 10, Lemma 3, scaled versions of Lemma 5 and 4 gives the statement in Lemma 1. □

We give the omitted proofs here.

**Lemma 6.** $\forall\{\boldsymbol{v}_j\}_{j=1}^n \in \mathcal{R}^d$, *triangular inequality satisfies*

$$\left\|\sum_{j=1}^n \boldsymbol{v}_j\right\|^2 \leq n\sum_{j=1}^n \|\boldsymbol{v}_j\|^2$$

*Proof.*

Using Jensen we get, $\left\|\frac{1}{n}\sum_{j=1}^n \boldsymbol{v}_j\right\|^2 \leq \frac{1}{n}\sum_{j=1}^n \|\boldsymbol{v}_j\|^2$. Multiplying both sides with $n^2$ gives the inequality. □

**Lemma 7.** *Algorithm 1 satisfies*

$$E\left\|\boldsymbol{h}^t\right\|^2 \leq C_t$$

*Proof.*

$$E \left\| \boldsymbol{h}^t \right\|^2 = E \left\| \frac{1}{m} \sum_{k \in [m]} \nabla L_k(\boldsymbol{\theta}_k^t) \right\|^2 = E \left\| \frac{1}{m} \sum_{k \in [m]} \left( \nabla L_k(\boldsymbol{\theta}_k^t) - \nabla L_k(\boldsymbol{\theta}_*) \right) \right\|^2$$

$$\leq \frac{1}{m} \sum_{k \in [m]} E \left\| \nabla L_k(\boldsymbol{\theta}_k^t) - \nabla L_k(\boldsymbol{\theta}_*) \right\|^2 = C_t$$

First equality is due to server update rule of $\boldsymbol{h}$ vector; second adds $(\nabla \ell(\boldsymbol{\theta}_*) = 0)$; third applies Jensen Inq.; and last one is the definition of $C_t$. $\square$

**Proof of Lemma 4**

$$\epsilon_t = \frac{1}{m} \sum_{k \in [m]} E \| \tilde{\boldsymbol{\theta}}_k^t - \boldsymbol{\gamma}^{t-1} \|^2 = \frac{1}{m} \sum_{k \in [m]} E \left\| \tilde{\boldsymbol{\theta}}_k^t - \boldsymbol{\theta}^{t-1} - \frac{1}{\alpha} \boldsymbol{h}^{t-1} \right\|^2$$

$$= \frac{1}{\alpha^2} \frac{1}{m} \sum_{k \in [m]} E \| \nabla L_k(\boldsymbol{\theta}_k^{t-1}) - \nabla L_k(\tilde{\boldsymbol{\theta}}_k^t) - \boldsymbol{h}^{t-1} \|^2$$

$$= \frac{1}{\alpha^2} \frac{1}{m} \sum_{k \in [m]} E \| \nabla L_k(\boldsymbol{\theta}_k^{t-1}) - \nabla L_k(\boldsymbol{\theta}_*) + \nabla L_k(\boldsymbol{\theta}_*) - \nabla L_k(\boldsymbol{\gamma}^{t-1}) + \nabla L_k(\boldsymbol{\gamma}^{t-1}) - \nabla L_k(\tilde{\boldsymbol{\theta}}_k^t) - \boldsymbol{h}^{t-1} \|^2$$

$$\leq \frac{4}{\alpha^2} \frac{1}{m} \sum_{k \in [m]} E \| \nabla L_k(\boldsymbol{\theta}_k^{t-1}) - \nabla L_k(\boldsymbol{\theta}_*) \|^2 + \frac{4}{\alpha^2} \frac{1}{m} \sum_{k \in [m]} E \| \nabla L_k(\boldsymbol{\gamma}^{t-1}) - \nabla L_k(\boldsymbol{\theta}_*) \|^2$$

$$+ \frac{4}{\alpha^2} \frac{1}{m} \sum_{k \in [m]} E \| \nabla L_k(\tilde{\boldsymbol{\theta}}_k^t) - \nabla L_k(\boldsymbol{\gamma}^{t-1}) \|^2 + \frac{4}{\alpha^2} E \| \boldsymbol{h}^{t-1} \|^2$$

$$\leq \frac{4}{\alpha^2} \frac{1}{m} \sum_{k \in [m]} E \| \nabla L_k(\boldsymbol{\theta}_k^{t-1}) - \nabla L_k(\boldsymbol{\theta}_*) \|^2 + \frac{4}{\alpha^2} \frac{1}{m} \sum_{k \in [m]} E \| \nabla L_k(\boldsymbol{\gamma}^{t-1}) - \nabla L_k(\boldsymbol{\theta}_*) \|^2$$

$$+ \frac{4}{\alpha^2} \frac{1}{m} \sum_{k \in [m]} E \| \nabla L_k(\tilde{\boldsymbol{\theta}}_k^t) - \nabla L_k(\boldsymbol{\gamma}^{t-1}) \|^2 + \frac{4}{\alpha^2} C_{t-1}$$

$$\leq \frac{8}{\alpha^2} C_{t-1} + \frac{4L^2}{\alpha^2} \epsilon_t + \frac{8L}{\alpha^2} E \left[ \ell(\boldsymbol{\gamma}^{t-1}) - \ell(\boldsymbol{\theta}_*) \right]$$

where first and second come from Eq. 9 and 8. Following inequalities come from Lemma 6, 7, smoothness and Inq. 5. Rearranging terms gives the Lemma. $\square$

**Proof of Lemma 5**

$$C_t = \frac{1}{m} \sum_{k \in [m]} E \| \nabla L_k(\boldsymbol{\theta}_k^t) - \nabla L_k(\boldsymbol{\theta}_*) \|^2$$

$$= \left( 1 - \frac{P}{m} \right) \frac{1}{m} \sum_{k \in [m]} E \| \nabla L_k(\boldsymbol{\theta}_k^{t-1}) - \nabla L_k(\boldsymbol{\theta}_*) \|^2 + \frac{P}{m} \frac{1}{m} \sum_{k \in [m]} E \| \nabla L_k(\tilde{\boldsymbol{\theta}}_k^t) - \nabla L_k(\boldsymbol{\theta}_*) \|^2$$

$$= \left( 1 - \frac{P}{m} \right) C_{t-1} + \frac{P}{m} \frac{1}{m} \sum_{k \in [m]} E \| \nabla L_k(\tilde{\boldsymbol{\theta}}_k^t) - \nabla L_k(\boldsymbol{\gamma}^{t-1}) + \nabla L_k(\boldsymbol{\gamma}^{t-1}) - \nabla L_k(\boldsymbol{\theta}_*) \|^2$$

$$\leq \left( 1 - \frac{P}{m} \right) C_{t-1} + \frac{2P}{m} \frac{1}{m} \sum_{k \in [m]} E \| \nabla L_k(\tilde{\boldsymbol{\theta}}_k^t) - \nabla L_k(\boldsymbol{\gamma}^{t-1}) \|^2$$

$$+ \frac{2P}{m} \frac{1}{m} \sum_{k \in [m]} E \| \nabla L_k(\boldsymbol{\gamma}^{t-1}) - \nabla L_k(\boldsymbol{\theta}_*) \|^2$$

$$\leq \left( 1 - \frac{P}{m} \right) C_{t-1} + \frac{2L^2 P}{m} \epsilon_t + \frac{2P}{m} \frac{1}{m} \sum_{k \in [m]} E \| \nabla L_k(\boldsymbol{\gamma}^{t-1}) - \nabla L_k(\boldsymbol{\theta}_*) \|^2$$

$$\leq \left( 1 - \frac{P}{m} \right) C_{t-1} + \frac{2L^2 P}{m} \epsilon_t + \frac{4LP}{m} E \left[ \ell(\boldsymbol{\gamma}^{t-1}) - \ell(\boldsymbol{\theta}_*) \right]$$

where first equality comes from taking expectation with respect to $\mathcal{P}_t$; second equality comes from definition of $C_t$. Inequalities follow from Lemma 6, smoothness and Inq. 5 respectively. $\square$

## B.2 STRONGLY CONVEX ANALYSIS

We state convergence for $\mu$ strongly convex and $L$ smooth $\{L_k\}_{k=1}^m$ functions as,

**Theorem 3.** *For $\mu$ strongly convex and $L$ smooth $\{L_k\}_{k=1}^m$ functions and $\alpha \geq \max\left(5\frac{m}{P}\mu, 30L\right)$, Algorithm 1 satisfies*

$$E\left[\ell\left(\frac{1}{R}\sum_{t=0}^{T-1}r^t\boldsymbol{\gamma}^t\right) - \ell(\boldsymbol{\theta}_*)\right] \leq \frac{1}{r^{T-1}}\left(20\alpha\left\|\boldsymbol{\theta}^0 - \boldsymbol{\theta}_*\right\|^2 + 400\frac{m}{P}\frac{1}{\alpha}\left(\frac{1}{m}\sum_{k\in[m]}\|\nabla L_k(\boldsymbol{\theta}_*)\|^2\right)\right)$$

*where* $\boldsymbol{\gamma}^t = \frac{1}{P}\sum_{k\in\mathcal{P}_t}\boldsymbol{\theta}_k^t$, $r = \left(1 + \frac{\mu}{\alpha}\right)$, $R = \sum_{t=0}^{T-1}r^t$, $\boldsymbol{\theta}_* = \arg\min_{\boldsymbol{\theta}}\ell(\boldsymbol{\theta})$.

If $\alpha = \max\left(5\frac{m}{P}\mu, 30L\right)$ we get the statement in Theorem 1. We will use the same $\{\tilde{\boldsymbol{\theta}}_k^t\}$, $\boldsymbol{\gamma}^t$, $C_t$, $\epsilon_t$ variables defined in Eq. 7, 8, 9.

With these definitions in mind, Theorem 3 can be seen as a direct consequence of the following Lemma,

**Lemma 8.** *For $\mu$ strongly convex and $L$ smooth $\{L_k\}_{k=1}^m$ functions, if $\alpha \geq \max\left(5\frac{m}{P}\mu, 30L\right)$, Algorithm 1 satisfies*

$$r\left(E\|\boldsymbol{\gamma}^t - \boldsymbol{\theta}_*\|^2 + \kappa C_t\right) \leq E\|\boldsymbol{\gamma}^{t-1} - \boldsymbol{\theta}_*\|^2 + \kappa C_{t-1} - \kappa_0 E\left[\ell(\boldsymbol{\gamma}^{t-1}) - \ell(\boldsymbol{\theta}_*)\right]$$

*where* $\kappa = \frac{8m(L+\alpha)}{z}$, $\kappa_0 = \frac{2\alpha^3 P + 2\alpha^2 P\mu - 2\alpha^2 m\mu - 40\alpha^2 LP - 80\alpha L^2 P - 40\alpha LP\mu + 8\alpha Lm\mu + 16L^2 m\mu - 80L^2 P\mu}{\alpha z}$,
$z = \alpha^3 P + \alpha^2 P\mu - \alpha^2 m\mu - 20\alpha L^2 P + 4L^2 m\mu - 20L^2 P\mu$, $r = \left(1 + \frac{\mu}{\alpha}\right)$.

Let's multiply Lemma 8 with $r^{t-1}$ and telescope as,

$$\kappa_0 r^{t-1}E\left[\ell(\boldsymbol{\gamma}^{t-1}) - \ell(\boldsymbol{\theta}_*)\right] \leq r^{t-1}\left(E\|\boldsymbol{\gamma}^{t-1} - \boldsymbol{\theta}_*\|^2 + \kappa C_{t-1}\right) - r^t\left(E\|\boldsymbol{\gamma}^t - \boldsymbol{\theta}_*\|^2 + \kappa C_t\right)$$

$$\kappa_0\sum_{t=1}^T r^{t-1}E\left[\ell(\boldsymbol{\gamma}^{t-1}) - \ell(\boldsymbol{\theta}_*)\right] \leq \left(E\|\boldsymbol{\gamma}^0 - \boldsymbol{\theta}_*\|^2 + \kappa C_0\right) - r^T\left(E\|\boldsymbol{\gamma}^T - \boldsymbol{\theta}_*\|^2 + \kappa C_T\right)$$

If $\alpha \geq \max\left(5\frac{m}{P}\mu, 30L\right)$, $\kappa_0$ and $\kappa$ become positive. Dividing both sides with $R = \sum_{t=0}^{T-1}r^t$ and eliminating negative terms on RHS gives,

$$\kappa_0\frac{1}{R}\sum_{t=1}^T r^{t-1}E\left[\ell(\boldsymbol{\gamma}^{t-1}) - \ell(\boldsymbol{\theta}_*)\right] \leq \frac{1}{R}\left(E\|\boldsymbol{\gamma}^0 - \boldsymbol{\theta}_*\|^2 + \kappa C_0\right)$$

Applying Jensen on LHS gives,

$$E\left[\ell\left(\frac{1}{R}\sum_{t=1}^T r^{t-1}\boldsymbol{\gamma}^{t-1}\right) - \ell(\boldsymbol{\theta}_*)\right] \leq \frac{1}{R}\frac{1}{\kappa_0}\left(\|\boldsymbol{\gamma}^0 - \boldsymbol{\theta}_*\|^2 + \kappa C_0\right)$$

We have $\frac{1}{R} = \frac{r-1}{r^T-1} \leq \frac{1}{r^{T-1}}$. Combining two inequalities, we get,

$$E\left[\ell\left(\frac{1}{R}\sum_{t=1}^T r^{t-1}\boldsymbol{\gamma}^{t-1}\right) - \ell(\boldsymbol{\theta}_*)\right] \leq \frac{1}{r^{T-1}}\frac{1}{\kappa_0}\left(\|\boldsymbol{\gamma}^0 - \boldsymbol{\theta}_*\|^2 + \kappa C_0\right)$$

which proves the statement in Theorem 3.

The proof of Lemma 8 is similar to the convex analysis. We generalize In. 6 to strongly convex functions for $\{L_k\}_{k=1}^m$s are $\mu$ strongly convex and $L$ smooth as,

$$-\langle\nabla L_k(\boldsymbol{x}), \boldsymbol{z} - \boldsymbol{y}\rangle \leq -L_k(\boldsymbol{z}) + L_k(\boldsymbol{y}) + \frac{L}{2}\|\boldsymbol{z} - \boldsymbol{x}\|^2 - \frac{\mu}{2}\|\boldsymbol{x} - \boldsymbol{y}\|^2 \quad \forall\boldsymbol{x}, \boldsymbol{y}, \boldsymbol{z} \qquad (11)$$

Since strongly convex functions are convex functions and we only change In. 6, we can directly use Lemma 2, 3, 4 and 5. Let's rewrite $\|\boldsymbol{\gamma}^t - \boldsymbol{\theta}_*\|^2$ expression as,

$$
\begin{aligned}
E\|\boldsymbol{\gamma}^t - \boldsymbol{\theta}_*\|^2 =& E\|\boldsymbol{\gamma}^{t-1} - \boldsymbol{\theta}_* + \boldsymbol{\gamma}^t - \boldsymbol{\gamma}^{t-1}\|^2 \\
=& E\|\boldsymbol{\gamma}^{t-1} - \boldsymbol{\theta}_*\|^2 + 2E\left[\left\langle\boldsymbol{\gamma}^{t-1} - \boldsymbol{\theta}_*, \boldsymbol{\gamma}^t - \boldsymbol{\gamma}^{t-1}\right\rangle\right] + E\|\boldsymbol{\gamma}^t - \boldsymbol{\gamma}^{t-1}\|^2 \\
=& E\|\boldsymbol{\gamma}^{t-1} - \boldsymbol{\theta}_*\|^2 + \frac{2}{\alpha m}\sum_{k\in[m]}E\left[\left\langle\boldsymbol{\gamma}^{t-1} - \boldsymbol{\theta}_*, -\nabla L_k(\tilde{\boldsymbol{\theta}}_k^t)\right\rangle\right] + E\|\boldsymbol{\gamma}^t - \boldsymbol{\gamma}^{t-1}\|^2 \\
\leq& \frac{2}{\alpha m}\sum_{k\in[m]}E\left[L_k(\boldsymbol{\theta}_*) - L_k(\boldsymbol{\gamma}^{t-1}) + \frac{L}{2}\|\tilde{\boldsymbol{\theta}}_k^t - \boldsymbol{\gamma}^{t-1}\|^2 - \frac{\mu}{2}\|\tilde{\boldsymbol{\theta}}_k^t - \boldsymbol{\theta}_*\|^2\right] \\
& + E\|\boldsymbol{\gamma}^{t-1} - \boldsymbol{\theta}_*\|^2 + E\|\boldsymbol{\gamma}^t - \boldsymbol{\gamma}^{t-1}\|^2 \\
=& E\|\boldsymbol{\gamma}^{t-1} - \boldsymbol{\theta}_*\|^2 - \frac{2}{\alpha}E\left[\ell(\boldsymbol{\gamma}^{t-1}) - \ell(\boldsymbol{\theta}_*)\right] + \frac{L}{\alpha}\epsilon_t - \frac{\mu}{\alpha}\frac{1}{m}\sum_{k\in[m]}E\|\tilde{\boldsymbol{\theta}}_k^t - \boldsymbol{\theta}_*\|^2 \\
& + E\|\boldsymbol{\gamma}^t - \boldsymbol{\gamma}^{t-1}\|^2 \\
\leq& E\|\boldsymbol{\gamma}^{t-1} - \boldsymbol{\theta}_*\|^2 - \frac{2}{\alpha}E\left[\ell(\boldsymbol{\gamma}^{t-1}) - \ell(\boldsymbol{\theta}_*)\right] + \frac{L}{\alpha}\epsilon_t - \frac{\mu}{\alpha}E\|\boldsymbol{\gamma}^t - \boldsymbol{\theta}_*\|^2 \\
& + E\|\boldsymbol{\gamma}^t - \boldsymbol{\gamma}^{t-1}\|^2
\end{aligned}
\tag{12}
$$

where we first expand the square term and use Lemma 2. Following inequalities use Inq. 11 and Lemma 9. Rearranging In. 12 gives,

$$
\left(1 + \frac{\mu}{\alpha}\right)E\|\boldsymbol{\gamma}^t - \boldsymbol{\theta}_*\|^2 \leq E\|\boldsymbol{\gamma}^{t-1} - \boldsymbol{\theta}_*\|^2 - \frac{2}{\alpha}E\left[\ell(\boldsymbol{\gamma}^{t-1}) - \ell(\boldsymbol{\theta}_*)\right] + \frac{L}{\alpha}\epsilon_t + E\|\boldsymbol{\gamma}^t - \boldsymbol{\gamma}^{t-1}\|^2
\tag{13}
$$

Let's define $z = \alpha^3 P + \alpha^2 P\mu - \alpha^2 m\mu - 20\alpha L^2 P + 4L^2 m\mu - 20L^2 P\mu$. Let's scale Lemma 4 and 5 with $\frac{\alpha(L+\alpha)(P\alpha + P\mu - m\mu)}{z}$ and $\frac{8m(L+\alpha)(\alpha+\mu)}{\alpha z}$ respectively. We note that the coefficients are positive due to the condition on $\alpha$. Summing Inq. 13, Lemma 3, scaled versions of Lemma 5 and 4 gives the statement in Lemma 8. $\square$

We give Lemma 9 and its proof here.

**Lemma 9.** *Algorithm 1 satisfies*

$$
-\frac{1}{m}\sum_{k\in[m]}E\|\tilde{\boldsymbol{\theta}}_k^t - \boldsymbol{\theta}_*\|^2 \leq -E\|\boldsymbol{\gamma}^t - \boldsymbol{\theta}_*\|^2
$$

*Proof.*

$$
\begin{aligned}
E\|\boldsymbol{\gamma}^t - \boldsymbol{\theta}_*\|^2 =& E\left\|\frac{1}{P}\sum_{k\in\mathcal{P}_t}(\boldsymbol{\theta}_k^t - \boldsymbol{\theta}_*)\right\|^2 \leq \frac{1}{P}E\left[\sum_{k\in\mathcal{P}_t}\|\boldsymbol{\theta}_k^t - \boldsymbol{\theta}_*\|^2\right] = \frac{1}{P}E\left[\sum_{k\in\mathcal{P}_t}\left\|\tilde{\boldsymbol{\theta}}_k^t - \boldsymbol{\theta}_*\right\|^2\right] \\
=& \frac{1}{m}\sum_{k\in[m]}E\left\|\tilde{\boldsymbol{\theta}}_k^t - \boldsymbol{\theta}_*\right\|^2
\end{aligned}
$$

where first equality comes from Eq. 9. The following inequality applies Jensen. Remaining relations are due to $\tilde{\boldsymbol{\theta}}_k^t = \boldsymbol{\theta}_k^t$ if $k \in \mathcal{P}_t$ and taking expectation by conditioning on randomness before time $t$. Rearranging the terms gives the statement in Lemma. $\square$

### B.3 NONCONVEX ANALYSIS

We state convergence for nonconvex $L$ smooth $\{L_k\}_{k=1}^m$s as,

**Theorem 4.** *For nonconvex and $L$ smooth $\{L_k\}_{k=1}^m$ functions and $\alpha \geq 20L\frac{m}{P}$, Algorithm 1 satisfies*

$$
E\left[\frac{1}{T}\sum_{t=1}^T\|\nabla\ell(\boldsymbol{\gamma}^{t-1})\|^2\right] \leq \frac{1}{T}\left(3\alpha\left(\ell(\boldsymbol{\theta}^0) - \ell_*\right) + 30L^3\frac{m}{P}\frac{1}{\alpha}\left(\frac{1}{m}\sum_{k\in[m]}E\|\boldsymbol{\theta}_k^0 - \boldsymbol{\theta}^0\|^2\right)\right) = O\left(\frac{1}{T}\right)
$$

*where $\boldsymbol{\gamma}^t = \frac{1}{P}\sum_{k\in\mathcal{P}_t}\boldsymbol{\theta}_k^t$, $\ell_* = \min_{\boldsymbol{\theta}}\ell(\boldsymbol{\theta})$.*

If $\alpha = 30L\frac{m}{P}$, we get the statement in Theorem 1. We will use $\{\tilde{\boldsymbol{\theta}}_k^t\}$ and $\boldsymbol{\gamma}^t$ variables as defined Eq. 7, 8, 9. Since we aim to find a stationary in the nonconvex case, let's define a new $C_t$ and keep $\epsilon_t$ the same as,

$$C_t = \frac{1}{m}\sum_{k\in[m]}E\|\boldsymbol{\theta}_k^t - \boldsymbol{\gamma}^t\|^2, \quad \epsilon_t = \frac{1}{m}\sum_{k\in[m]}E\|\tilde{\boldsymbol{\theta}}_k^t - \boldsymbol{\gamma}^{t-1}\|^2$$

Similarly, $C_t$ tracks how well local models approximate the current active device average. Upon convergence $C_t$ and $\epsilon_t$ will be 0.

Theorem 4 can be seen as a direct consequence of the following Lemma,

**Lemma 10.** *For $L$ smooth $\{L_k\}_{k=1}^m$ functions, if $\alpha \geq 20L\frac{m}{P}$, Algorithm 1 satisfies*

$$E\left[\ell(\boldsymbol{\gamma}^t)\right] + \kappa C_t \leq E\left[\ell(\boldsymbol{\gamma}^{t-1})\right] + \kappa C_{t-1} - \kappa_0 E\left\|\nabla\ell(\boldsymbol{\gamma}^{t-1})\right\|^2$$

*where $\kappa = 4L^3P\frac{\alpha+L}{\alpha}\frac{2m-P}{z}, \kappa_0 = \frac{1}{2\alpha}\frac{\alpha^2P^2-4\alpha LP^2-32L^2m^2-16L^2Pm-24L^2P^2}{z},$*
*$z = \alpha^2P^2 - 32L^2m^2 + 16L^2Pm - 20L^2P^2.$*

Lemma 10 can be telescoped as,

$$\kappa_0 E\left\|\nabla\ell(\boldsymbol{\gamma}^{t-1})\right\|^2 \leq \left(E\left[\ell(\boldsymbol{\gamma}^{t-1})\right] - \ell_* + \kappa C_{t-1}\right) - \left(E\left[\ell(\boldsymbol{\gamma}^t)\right] - \ell_* + \kappa C_t\right)$$

$$\kappa_0 \sum_{t=1}^T E\left\|\nabla\ell(\boldsymbol{\gamma}^{t-1})\right\|^2 \leq \left(E\left[\ell(\boldsymbol{\gamma}^0)\right] - \ell_* + \kappa C_0\right) - \left(E\left[\ell(\boldsymbol{\gamma}^T)\right] - \ell_* + \kappa C_T\right)$$

If $\alpha \geq 20L\frac{m}{P}$, we have $\kappa_0$ and $\kappa$ as positive quantities. By definition, we also have $C_t$ sequences as positive. Eliminating negative terms on RHS and summing over time give,

$$E\left[\frac{1}{T}\sum_{t=1}^T\|\nabla\ell(\boldsymbol{\gamma}^{t-1})\|^2\right] \leq \frac{1}{T}\frac{1}{\kappa_0}\left(\ell(\boldsymbol{\theta}^0) - \ell_* + \kappa\left(\frac{1}{m}\sum_{k\in[m]}E\|\boldsymbol{\theta}_k^0 - \boldsymbol{\theta}^0\|^2\right)\right)$$

which proves the statement in Theorem 2.

The proof of Lemma 10 builds on Inq. 4 where we upper bound $\ell(\boldsymbol{\gamma}^t)$ with $\ell(\boldsymbol{\gamma}^{t-1})$. Inq. 4 gives $(\boldsymbol{\gamma}^t - \boldsymbol{\gamma}^{t-1})$ and $\nabla\ell(\boldsymbol{\gamma}^{t-1})$ on RHS. We state a set of Lemmas to tackle these terms. We note here that Lemma 2 and 3 holds since $\epsilon_t$ is the same as in convex case.

To bound excess $\epsilon_t$ term, we introduce two more Lemmas as

**Lemma 11.** *For $L$ smooth $\{L_k\}_{k=1}^m$ functions, Algorithm 1 satisfies*

$$\left(1 - 4L^2\frac{1}{\alpha^2}\right)\epsilon_t \leq 8L^2\frac{1}{\alpha^2}C_{t-1} + 4\frac{1}{\alpha^2}E\left\|\nabla\ell(\boldsymbol{\gamma}^{t-1})\right\|^2$$

**Lemma 12.** *For $L$ smooth $\{L_k\}_{k=1}^m$ functions, Algorithm 1 satisfies*

$$C_t \leq 2\frac{m-P}{2m-P}C_{t-1} + 2\frac{P}{2m-P}\epsilon_t + 2\frac{m}{P}E\left\|\boldsymbol{\gamma}^t - \boldsymbol{\gamma}^{t-1}\right\|^2$$

Using Inq. 4 we get,

$$E\left[\ell(\boldsymbol{\gamma}^t)\right] - E\left[\ell(\boldsymbol{\gamma}^{t-1})\right] - \frac{L}{2}E\|\boldsymbol{\gamma}^t - \boldsymbol{\gamma}^{t-1}\|^2 \leq E\left[\langle\nabla\ell(\boldsymbol{\gamma}^{t-1}), \boldsymbol{\gamma}^t - \boldsymbol{\gamma}^{t-1}\rangle\right]$$

$$= \frac{1}{\alpha}E\left[\left\langle\nabla\ell(\boldsymbol{\gamma}^{t-1}), \frac{1}{m}\sum_{k\in[m]}-\nabla L_k(\tilde{\boldsymbol{\theta}}_k^t)\right\rangle\right]$$

$$\leq \frac{1}{2\alpha}E\left\|\frac{1}{m}\sum_{k\in[m]}\left(\nabla L_k(\tilde{\boldsymbol{\theta}}_k^t) - \nabla L_k(\boldsymbol{\gamma}^{t-1})\right)\right\|^2 - \frac{1}{2\alpha}E\left\|\nabla\ell(\boldsymbol{\gamma}^{t-1})\right\|^2$$

$$\leq \frac{1}{2\alpha}\frac{1}{m}\sum_{k\in[m]}E\left\|\nabla L_k(\tilde{\boldsymbol{\theta}}_k^t) - \nabla L_k(\boldsymbol{\gamma}^{t-1})\right\|^2 - \frac{1}{2\alpha}E\left\|\nabla\ell(\boldsymbol{\gamma}^{t-1})\right\|^2$$

$$\leq \frac{L^2}{2\alpha}\epsilon_t - \frac{1}{2\alpha}E\left\|\nabla\ell(\boldsymbol{\gamma}^{t-1})\right\|^2 \tag{14}$$

where first equality uses Lemma 2. The following inequalities are due to $\langle \boldsymbol{a}, \boldsymbol{b}\rangle \leq \frac{1}{2}\|\boldsymbol{b}+\boldsymbol{a}\|^2 - \frac{1}{2}\|\boldsymbol{a}\|^2$, Jensen Inq. and smoothness.

Let's define $z = \alpha^2 P^2 - 32L^2m^2 + 16L^2Pm - 20L^2P^2$ and scale Lemma 12, 3 and 11 with $z_0 = 4L^3P\frac{\alpha+L}{\alpha}\frac{2m-P}{z}$,
$z_1 = \frac{L}{2} + z_0\frac{2m}{P}$, and $z_2 = LP^2\frac{\alpha}{2}\frac{L+\alpha}{z}$ respectively. We note that the coefficients are positive due to the condition on $\alpha$. Summing Inq. 14, scaled versions of Lemma 3, 11 and 12 gives the statement in Lemma 10. $\square$

Lastly, we note that the convergence analysis is given with respect to L2 norm in the gradients. L2 norm arises in the analysis because In. 4 has L2 norm due to our definition of smoothness. Furthermore, the analysis can be extended to different norms. To do so, smoothness needs to be defined with respect to primal and dual norms as in Eq. 3 in Nesterov et al. (2020).

We give the omitted proofs here.

**Proof of Lemma 11**

$$\epsilon_t = \frac{1}{m}\sum_{k\in[m]}E\|\tilde{\boldsymbol{\theta}}_k^t - \boldsymbol{\gamma}^{t-1}\|^2 = \frac{1}{m}\sum_{k\in[m]}E\left\|\tilde{\boldsymbol{\theta}}_k^t - \boldsymbol{\theta}^{t-1} - \frac{1}{\alpha}\boldsymbol{h}^{t-1}\right\|^2$$

$$= \frac{1}{\alpha^2}\frac{1}{m}\sum_{k\in[m]}E\|\nabla L_k(\boldsymbol{\theta}_k^{t-1}) - \nabla L_k(\tilde{\boldsymbol{\theta}}_k^t) - \boldsymbol{h}^{t-1}\|^2$$

$$= \frac{1}{\alpha^2}\frac{1}{m}\sum_{k\in[m]}E\|\nabla L_k(\boldsymbol{\theta}_k^{t-1}) - \nabla L_k(\boldsymbol{\gamma}^{t-1}) + \nabla L_k(\boldsymbol{\gamma}^{t-1}) - \nabla L_k(\tilde{\boldsymbol{\theta}}_k^t) - \nabla\ell(\boldsymbol{\gamma}^{t-1}) + \nabla\ell(\boldsymbol{\gamma}^{t-1}) - \boldsymbol{h}^{t-1}\|^2$$

$$\leq \frac{4}{\alpha^2}\frac{1}{m}\sum_{k\in[m]}E\|\nabla L_k(\boldsymbol{\theta}_k^{t-1}) - \nabla L_k(\boldsymbol{\gamma}^{t-1})\|^2 + \frac{4}{\alpha^2}\frac{1}{m}\sum_{k\in[m]}E\|\nabla L_k(\boldsymbol{\gamma}^{t-1}) - \nabla L_k(\tilde{\boldsymbol{\theta}}_k^t)\|^2$$

$$+ \frac{4}{\alpha^2}E\|\nabla\ell(\boldsymbol{\gamma}^{t-1})\|^2 + \frac{4}{\alpha^2}E\|\nabla\ell(\boldsymbol{\gamma}^{t-1}) - \boldsymbol{h}^{t-1}\|^2$$

$$\leq \frac{4}{\alpha^2}\frac{1}{m}\sum_{k\in[m]}E\|\nabla L_k(\boldsymbol{\theta}_k^{t-1}) - \nabla L_k(\boldsymbol{\gamma}^{t-1})\|^2 + \frac{4}{\alpha^2}\frac{1}{m}\sum_{k\in[m]}E\|\nabla L_k(\boldsymbol{\gamma}^{t-1}) - \nabla L_k(\tilde{\boldsymbol{\theta}}_k^t)\|^2$$

$$+ \frac{4}{\alpha^2}E\|\nabla\ell(\boldsymbol{\gamma}^{t-1})\|^2 + \frac{4}{\alpha^2}\frac{1}{m}\sum_{k\in[m]}E\|\nabla L_k(\boldsymbol{\gamma}^{t-1}) - \nabla L_k(\boldsymbol{\theta}_k^{t-1})\|^2$$

$$\leq \frac{8L^2}{\alpha^2}C_{t-1} + \frac{4L^2}{\alpha^2}\epsilon_t + \frac{4}{\alpha^2}E\|\nabla\ell(\boldsymbol{\gamma}^{t-1})\|^2$$

where first, second and third come from definition of $\epsilon_t$, Eq. 9 and 8. The following inequalities are due to Lemma 6, Jensen Inq. and smoothness. Rearranging terms gives the Lemma. $\square$

**Proof of Lemma 12**

$$C_t = \frac{1}{m}\sum_{k\in[m]}E\|\boldsymbol{\theta}_k^t - \boldsymbol{\gamma}^t\|^2 = \frac{1}{m}\sum_{k\in[m]}E\|\boldsymbol{\theta}_k^t - \boldsymbol{\gamma}^{t-1} + \boldsymbol{\gamma}^{t-1} - \boldsymbol{\gamma}^t\|^2$$

$$\leq \left(1 + \frac{P}{2m-P}\right)\frac{1}{m}\sum_{k\in[m]}E\|\boldsymbol{\theta}_k^t - \boldsymbol{\gamma}^{t-1}\|^2 + \left(1 + \frac{2m-P}{P}\right)\frac{1}{m}\sum_{k\in[m]}E\|\boldsymbol{\gamma}^t - \boldsymbol{\gamma}^{t-1}\|^2$$

$$= \frac{P}{m}\left(1 + \frac{P}{2m-P}\right)\frac{1}{m}\sum_{k\in[m]}E\|\tilde{\boldsymbol{\theta}}_k^t - \boldsymbol{\gamma}^{t-1}\|^2$$

$$+ \left(1 - \frac{P}{m}\right)\left(1 + \frac{P}{2m-P}\right)\frac{1}{m}\sum_{k\in[m]}E\|\boldsymbol{\theta}_k^{t-1} - \boldsymbol{\gamma}^{t-1}\|^2 + \left(1 + \frac{2m-P}{P}\right)E\|\boldsymbol{\gamma}^t - \boldsymbol{\gamma}^{t-1}\|^2$$

$$= \frac{P}{m}\left(1 + \frac{P}{2m-P}\right)\epsilon_t + \left(1 - \frac{P}{m}\right)\left(1 + \frac{P}{2m-P}\right)C_{t-1} + \left(1 + \frac{2m-P}{P}\right)E\|\boldsymbol{\gamma}^t - \boldsymbol{\gamma}^{t-1}\|^2$$

where we start with definition of $C_t$. First inequality is due to $\|\boldsymbol{a} + \boldsymbol{b}\|^2 \leq (1+z)\|\boldsymbol{a}\|^2 + \left(1 + \frac{1}{z}\right)\|\boldsymbol{b}\|^2$ for $z > 0$. The following equality takes expectation conditioned on randomness before time $t$. Since each device is selected with probability $\frac{P}{m}$, $\boldsymbol{\theta}_k^t$ is a random variable that is equal to $\tilde{\boldsymbol{\theta}}_k^t$ with probability $\frac{P}{m}$. Otherwise, it is $\boldsymbol{\theta}_k^{t-1}$. Final equality is due to definitions of $\epsilon_t$ and $C_t$. $\square$

