# OpenReview forum: "Federated Learning Based on Dynamic Regularization"
_ICLR.cc/2021/Conference — ICLR 2021 Oral_

### Official Review · AnonReviewer3 · 2020-10-24
**It is a good paper but needs more work.**

**Rating:** 8
**Confidence:** 4

**Review:**

The paper proposes a new optimization method named FedDyn to handle data heterogeneity inherent in FL via a dynamic regularization. Such a dynamic regularization modifies each local objective by adding a linear and quadratic term that makes the local stationary point is asymptotically consistent with that of global objectives. The authors prove convergence results for FedDyn in both general convex and non-convex cases and test FedDyn on synthetic and real-world datasets.

Pros:
1. The paper is well written and easy to follow. All proof seems correct.
2. The experiment setup follows many previous works, which facilitate comparisons, and I appreciate a lot. Besides, a lot of details are given, which helps reproduction.
3. The proposed method has great superiority over other baselines in communication efficiency, shown by a lot of experiment results.

Cons:
1. I think the author could also analyze FedDyn in a strongly-convex case since in that well-conditioned case the convergence performance of FedDyn could give more insights for researchers. I expect FedDyn would converge to the optima with a rate exponential in communication rounds.
2. The author declare that FedDyn has great advantages over competing methods in cases of a large number of devices. However, from the main theorem, I couldn't figure out the reason.
3. The author declare that FedDyn is more robust to unbalanced data than baseline methods. From experiments, it seems correct as  FedDyn achieves a larger factor of gains over others in unbalanced data. However, again, no theoretical analysis is given. The main theorem is about the convergence of balance data in terms of communication rounds not about the effect of unbalance data.
4. Since the proof mainly follows from that of SCAFFOLD and many empirical findings have no theoretical support, I would regard the main contribution as the algorithm itself. The experiment indeed did well, however, there is still some shortcomings. There are other algorithms proposed to (or able to) handle data heterogeneity like FedPD [1], FedSplit [2] and VRL-SGD [3]. However, the authors didn’t consider these competing methods and even didn’t mention them.

[1] Zhang, Xinwei, et al. "FedPD: A Federated Learning Framework with Optimal Rates and Adaptivity to Non-IID Data." arXiv preprint arXiv:2005.11418 (2020).
[2] Pathak R, Wainwright M J. FedSplit: An algorithmic framework for fast federated optimization[J]. arXiv preprint arXiv:2005.05238, 2020.
[3] Liang, Xianfeng, et al. "Variance reduced local SGD with lower communication complexity." arXiv preprint arXiv:1912.12844 (2019).

Other points:
1. The description of SCAFFOLD’s linear term seems imprecise. The subtracted term is not $\frac{1}{m} \sum_{k \in[m]} \nabla L_{k}\left(\boldsymbol{\theta}_{k}^{t}\right)$, since SCAFFOLD only activates a small fraction of devices and this term is impossible to compute. But I can understand what the author wants to convey: their affine function saves communication costs.
2. The authors mention that at each communication round SCAFFOLD communicates the current model and its associated gradient while others communicate only the former. So, if I want to say FedDyn is more effective in saving communication rounds, I would expect it to achieve at least 2x gains over SCAFFOLD. However, this is a rare case when the participation rate is 10% (see Table 2). It seems that FedDyn is not so effective in saving communication rounds and the main reason for its effectiveness seems to be that the modified affine function requires once communication per round.


--------------------------------------------------------------------
I have read the authors' responses and almost all my concerns have been well addressed.
So I increase my point to 8.

---

> ### Author Response · Authors · 2020-11-18
> **We show convergence for strongly convex functions, give theoretical reasoning of the advantage for large number of devices and unbalanced data distributions, present differences compared to the cited references, and discuss effectiveness of FedDyn in terms of communication rounds.**
>
> Q1: Could we also prove convergence for strongly convex functions?
>
> *Yes. FedDyn, in fact, achieves linear rate for strongly convex functions which is a significant improvement over convex rate. We refer to the revised draft (see Theorem 1 and Appendix B.2. for analysis.).*
>
> Q2: How can we infer from Theorem 1 that FedDyn handles massive devices?
>
> *FedDyn convergence rate is invariant to number of devices $m$, and depends only on the participation ratio, $m/P$, where $P$ is the number of active devices at any time.* Our reported experimental results examine accuracy vs. communication for different participation levels and different $m$. Existing works in this context typically report $m/P = 10, m=100$. We consider $m=1000$ as well (see Table 1).
>
> In theory, it is a useful exercise to see how accuracy or required communication rounds change with number of devices, while keeping $P$ fixed. We see, for instance, in the convex case, we exhibit a convergence rate $O\left(\frac{1}{T}\sqrt{\frac{m}{P}}\right)$. This implies that as we increase the number of devices, while keeping $P$ fixed, the additional number of communications scales sub-linearly ($T = O(\sqrt{m})$) to maintain the same level of accuracy. Evidently, this is significantly better than Scaffold who require $T = O(m)$. We refer to paragraph starting with 'Dynamic Regularization:' on pg. 2 in the revised paper for more details.
>
> Q3: How can we infer from Theorem 1, FedDyn's performance with Unbalanced data?
>
> *Response:* As it stands, Theorem 1 is stated for the balanced data setting, where the number of data points for each device $j$ is $N_j=N/m$ for a total of $N$ data points across all $m$ devices. Nevertheless, our proof directly extends to the unbalanced case. All we need is to re-define device-level and global objectives, so that they align with the desired empirical loss, $\ell(\theta)$, namely, the average loss across all data instances.
> $$
> \ell(\theta) = \frac{1}{N}\sum_{j\in[m], i\in[N_j]}L(z_i^j;\theta)
> $$
> where, $z_i^j$ is the ith data sample in the jth device; $L(z_i^j;\theta)$ is the loss observed for model $\theta$ on this sample instance. Observe that the loss for a particular instance depends only on the model $\theta$ and not the device, and as such represents the empirical average over all data points in the federated system.
>
> To this end, consider the case where $N_j$'s are not equal but arbitrary positive integers such that $\sum_{j \in [m]}N_j=N$. Let us re-define the device loss as:
> $$L_j(\theta) = \frac{m}{N}\sum_{i\in[N_j]}L(z_i^j; \theta),$$
> and the cummulative loss, $\ell'(\theta)$ as the empirical average of device losses:
> $$
> \ell'(\theta)= \frac{1}{m} \sum_{j \in [m]} L_j(\theta).
> $$
> It is straightforward to verify that,
> $$\ell'(\theta) = \frac{1}{m}\sum_{j\in[m]}L_j(\theta)=\frac{1}{m}\sum_{j\in[m]}\frac{m}{N}\sum_{i\in[N_j]}L(z_i^j; \theta)=\frac{1}{N}\sum_{j\in[m], i\in[N_j]}L(z_i^j;\theta)=\ell(\theta).$$
> We can check that all of the steps in the proof of Theorem 1 continues to hold, and as such this implies that the statement of Theorem 1 holds for unbalanced setting with no additional modifications.
>
> On the other hand, we caution the reader that the convergence rate here and in prior works is in expectation, and a stronger notion such as in probability or mean-squared sense, maybe desirable.

---

> > ### Author Response · Authors · 2020-11-18
> > **Continued**
> >
> > Q4: What is the relationship of FedDyn to recent works (FedSplit, FedPD, VRL-SGD)?
> >
> > We were unaware of these works, and thank the reviewer for pointing us to them. We cite them in our revised draft. Let us list the key differences:
> >
> > *FedDyn handles partial participation, while these recent works, in essence, require full participation.* In passing we emphasize that partial participation, where devices are activated in a random fashion is of fundamental practical importance in federated learning.
> >
> > We refer to our detailed response to reviewer 1 for FedSplit comparison. VRL-SGD also requires full participation. On the other hand, FedPD appears to allow for partial participation. But the nature of partial participation is such that all devices are *simultaneously* active or not with some probability. As such, this is a strong assumption, which is not met in practice (think of massive number of devices). Note that the existing full-participation results in ours and prior works directly apply to this case by trivially freezing updates when inactive. Nevertheless, the interesting situation is to update models even when no device is active. We agree that this could be interesting but somewhat outside the scope of our framework.
> >
> > In summary, like we point out for FedSplit, partial participation, as in our framework, poses fundamental challenges in theory, and in practice reasonable modifications do not appear to yield good performance.
> >
> > *There are also important conceptual and operational differences even for the full-participation setting.* We refer to our Reviewer 1 response for FedSplit.
> >
> > VRL-SGD is a SGD type method, where the goal of the server, in essence, is to leverage devices to produce gradient information. In its extreme, an SGD method would choose a device at random, and the device performs a few SGD steps, and return the resulting gradient as output to the server. In contrast, our devices serve as full optimizers, and as such, we are agnostic to as to how to solve the optimization problem. In point of fact, in practice, we use SGD to optimize device objectives as a matter of computational convenience, and as evident the number of such SGD steps could be as large as $10^5$, a number so large that it ceases to be an SGD type method! On the other hand, Theorem 5.1 in VRL-SGD paper states its convergence rate where $k$ is the number SGD updates, $\gamma$ is the learning rate and the product $k\gamma$ is upper bounded with a problem dependent constant. Due to the bound on $k\gamma$, if we allow more SGD steps, we need to decrease $\gamma$. But, this we believe in turn negatively impacts convergence. Evidently, VRL-SGD accuracy degrades with more SGD steps, while FedDyn is agnostic to it so long as we reach a stationary point.
> >
> > We further refer to our response to Reviewer 4 to highlight other differences between SGD methods and ours. As pointed out in our response to Reviewer 1, an important aspect is that our gradient updates bear a linear relationship (a consequence of optimality condition), which is critical for our ease of analysis in the partial participation setting. VRL-SGD updates are somewhat more implicit, and it is not clear whether it is easy to extend their analysis to partial participation. Although FedPD is not an SGD-type algorithm, their updates bear an implicit relationship between the model and gradient (like FedSplit), which we believe is somewhat difficult to extend in the randomly active (our partial participation) scenario.
> >
> > To summarize, FedDyn and the cited works exhibit differences in terms of scope and operation. We test FedSplit among these works and empirical results reveals Scaffold and FedDyn outperform FedSplit for both full and partial participation levels (see Appendix A.4).

---

> > > ### Author Response · Authors · 2020-11-18
> > > **Continued**
> > >
> > > Q5: Description of SCAFFOLD's linear term,
> > >
> > > We agree that the statement does not hold for partial participation case. We added a note indicating the statement is for full participation setting.
> > >
> > > Q6: How effective is FedDyn in terms of communication rounds?
> > >
> > > *Our work aims to decrease energy costs in Federated Learning problem.* As stated in Introduction, communication is the main source of energy consumption (see our response to Rev. 2 for more details). Since we view Federated Learning as a practical problem, we aim to decrease both the amount of bits transmitted as well as communication rounds. We agree that communication round is different from reported quantities in Tables, and will clarify it further in revised version.
> > >
> > > *Cifar10 and Cifar100 results are better than MNIST and EMNIST-L.* We note that Cifar10 and Cifar100 results show more than $2\times$ gain over SCAFFOLD in 10\% participation setting for the top target accuracy levels. These results are in the block entitled Moderate in Table 1 and in Table 2. We believe that Cifar10 and Cifar100 results represent a more complex setting than MNIST and EMNIST. As such top accuracy in MNIST is somewhat easier to achieve, and so MNIST results are less reflective of the difficulty of issues posed in federated learning.
> > >
> > > *From a practical perspective, sending two models requires twice the bandwidth, and number of bits transmitted is a practical concern.* In mobile wireless scenarios, the quality of channel changes continuously. As such, we can encounter situations where only a fraction of the message is successfully transmitted. Naively, one can just ignore such devices and assume that active devices are the ones that successfully transmitted the device model and the gradient. However, this would decrease the participation ratio and increase number of rounds required.
> > >
> > > To summarize, our work views Federated Learning from a practical perspective and aims to decrease communication costs both in number of rounds and bits/round. Our experiments for Cifar10 and Cifar100 indeed shows more than $2\times$ gain.

---

### Official Review · AnonReviewer4 · 2020-10-27
**Interesting new algorithm for distributed optimization / federated learning**

**Rating:** 7
**Confidence:** 5

**Review:**

* Context: the authors propose a new distributed optimization algorithm, called FedDyn, to minimize a sum of smooth functions. Their motivation is the context of federated learning, in which each function is the loss of one user corresponding to its data stored locally. The goal is to reduce the communication burden to achieve the true global solution (not an approximation thereof) to a given accuracy.

* Strengths:
1. The main strength of the algorithm is its robustness to partial collaboration, in which only a subset of the devices participate at every iteration. Indeed, in case of full participation, it is not clear that the algorithm has any advantage in comparison with classical (S)GD-type methods.
2. The experiments are extensive, well described, and convincing: the method achieves the goal of convergence to a given accuracy with a speed and total communication load competitive w.r.t other methods.

* Weaknesses:
The comparison to methods based on local steps, like FedAvg and Scaffold, is somewhat unnatural: these methods go along the idea of doing more computations between communication rounds. This is not the case of the proposed method, which is much closer in spirit to the class of SGD methods. Thus, I find the discussion, which turns around 'correcting' the drawbacks of  methods based on local steps, in particular the fact that FedAvg converges to an approximate solution, convoluted.  Moreover,
1. No regularizer at the master in the objective function.
2. The local loss functions must be smooth and in addition, their proximity operators must be computable (see point 5. below).
3. No comparison to accelerated SGD-type method with accuracy in O(1/T^2) instead of O(1/T).
4. No discussion of possible linear convergence in case of strong convexity.
5. alpha, the inverse of the stepsize, must be large (>25L) for Theorem 1 to apply. This shows that the analysis is not tight at all.

* assessment: I think the paper deserves publication, since the proposed algorithm is new, comes with some convergence guarantees, and shows good performance in practice. However, I view the theoretical analysis as a preliminary one, and several aspects would deserve to be investigated more in depth.

More detailed discussion:

1. There should be a discussion about the literature of SGD type methods. Indeed, partial participation takes the weak form, in Theorem 1, of a subset selected uniformly at random. Some papers by Richtarik et al coming to my mind: "A Unified Theory of SGD: Variance Reduction, Sampling, Quantization and Coordinate Descent", "Unified analysis of stochastic gradient methods for composite convex and smooth optimization", "A unified analysis of stochastic gradient methods for nonconvex federated optimization".

2. You should talk, even very shortly, about the other strategy to decrease the communication burden: compression, see for instance the recent papers and refs therein: "On the Discrepancy between the Theoretical Analysis and Practical Implementations of Compressed Communication for Distributed Deep Learning", "Distributed learning with compressed gradient differences".

3. Further on, one could view the proposed method as GD with unbiased compression: all devices compute their new gradient/model but then with probability P/m it is sent to the master, otherwise it is not sent. It would be good to investigate this relationship more closely.

4. When mentioning prior work on methods using local steps of SGD, which is "inconsistent with minimizing the global loss" (or later when mentioning that "performance degrades in non-IID scenarios"), you can cite the paper "From Local SGD to Local Fixed Point Methods for Federated Learning", ICML 2020, which gives a precise characterization of this "inconsistency" in the strongly convex case (Theorem 2.14). For the non-strongly convex case, you can refer to the paper "Tighter theory for local SGD on identical and heterogeneous data".

5. Each local computation step is actually a call to the proximity operator of L_k: we have theta_k^t = prox_{L_k/alpha}(theta^{t-1}+grad.L_k(theta_k^{t-1})/alpha). This should be mentioned clearly, as well as the assumption that L_k is proximable (in addition of being smooth). The operation is not a proximal gradient descent step, since there is a + and not a - in front of the gradient. Still, the variable h has the flavor of a dual variable. So, the relationship to proximal splitting algorithms should be investigated, since there seems to be a connection there.
Thinking a bit about this connection, I found out the "distributed Davis-Yin algorithm" in the paper "Distributed Proximal Splitting Algorithms
with Rates and Acceleration", see p. 22 of https://arxiv.org/pdf/2010.00952.pdf. Rewritten in your context, its iteration is:
|for each device in parallel do
|	theta_k^t = prox_{L_k/alpha}(2.theta^{t-1}-s_k^{t-1}
|		-grad.L_k(theta^{t-1})/alpha)
|	s_k^t = s_k^{t-1} + theta_k^t - theta^{t-1}
|	transmit theta_k^t to server
|end for
|s^t = s^{t-1} +  1/m.sum_k (theta_k^t - theta^{t-1})
|theta^t = s^t
There seem to be several differences between this algorithm and yours, but still, they look similar in spirit. It would be very interesting to compare them.

6. In the last two steps of your algorithm (set h^t=..., set theta^t=...) you can remove the multiplication and division by alpha.

7. For the nonconvex case, the rate in Theorem 1 is with respect to the gradient norm. You should tell a bit about the literature, see e.g the discussion in "Primal-dual accelerated gradient descent with line search for convex and nonconvex optimization problems" by Nesterov et al.

Typos:
* convergences -> converges
* non convex -> nonconvex
* Table 1: I guess the "Acc." column is to provide the accuracy. This should be said, since the reader might be confused and think that this is an accelerated method included in the comparison
* theta_k^infty triangle theta^infty: what does the triangle mean?

---

> ### Author Response · Authors · 2020-11-12
> **We differentiate FedDyn from SGD based mehtods, show convergence for strongly convex functions, present differences compared to Distributed Davis-Yin method and the proximal splitting methods.**
>
> Q1: Is FedDyn more close to SGD type methods than the baselines such as Scaffold? Is it basically an unbiased gradient descent method? Why don't we compare FedDyn to accelerated SGD type methods? There are missing discussion on SGD type methods,
>
> *No. FedDyn is not an SGD type method.* Essentially, in many SGD type methods (such as VRL-SGD or accelerated SGD), the server transmits current model to devices, and devices perform SGD with device loss, usually for a few rounds, and transmit the result to the server.
>
> In contrast, we are agnostic to SGD, using it merely as a computational tool in our experiments, and our objective (as noted by Reviewer 1 & Reviewer 2) is really to solve device objective to optima (or reaching a stationary solution).
>
> In particular, for our theory to work out, we assume (see Eq. 2 on pg. 4) first-order conditions are fully satisfied. As such, we may realize this condition either using SGD, that runs for a large number of epochs, or a off-the-shelf optimization solver.
>
> To baseline our approach against existing works such as Scaffold, who like us are not of SGD-type, and SGD over mini-batches and many epochs, we used SGD and ran it over similar number of mini-batches and epochs to benchmark against similar computational times (see Appendix A.2). Recall each epoch in essence runs SGD over entire dataset in the device, and we run many epochs. As such our setup does not resembles SGD-type at all. Surprisingly, as noted in our R1 response, in practice, these mini-batches and number of epochs is often sufficient to reach close to a stationary solution in our experiments.
>
>
> *For SGD methods, more epochs hurt performance, whereas more SGD steps only helps improve our performance.* Another way to see why our method is not SGD type, is that the SGD type methods, implicitly or explicitly require few SGD rounds in a device so that the model updates do not drift too far from the centralized SGD updates. For example, see our response to Reviewer 3 regarding VRL-SGD where more SGD steps decreases the accuracy. On the other hand, our overall framework is agnostic to SGD so long as stationarity condition is satisfied for the device. Furthermore, more SGD steps helps, as it would reduce the error between solution we reach and the desired stationary solution.
>
> *The cited SGD type references require full participation settings.* While we were unaware of these works, and will cite them our revision, we point out yet another aspect that is a fundamental issue for FL. As stated in our introduction, we focus on four fundamental aspects of Federated Learning which are partial device participation (communication), heterogeneity, data imbalance, and massive device levels. Accelerated SGD type methods such as FedAc (Yuan \& Ma,2020) and Accelerated DIANA (Li et al., 2020c), and other cited references that unify SGD analysis (Gorbunov et al., 2020; Khaled et al., 2020b) are proposed for full participation. Partial participation is of a significant concern in practice since we will not be in a case where we have all devices available at each round. Moreover, adapting a full-participation method to a partial participation does not appear to be straightforward (see FedSplit discussion in response to Reviewer 1). Therefore, we do not see a good reason to empirically compare our algorithm to these methods.
>
> *FedDyn supports partial participation.* The stationary point relation of device level objectives allows FedDyn to linearly relate the gradient from previous round to the current round as described in Eq. 2 (Pg. 4). Different from SGD type methods, this relation significantly simplifies the analysis and makes it easier to extend to the partial settings.
>
> *We do not view FedDyn as SGD, and furthermore, we do not see it as an unbiased gradient descent method.* In SGD the expectation over data is the gradient of the loss across all data. For FedDyn, a device looks through all the data (large number of epochs), whenever it is active, and as such we do not see how to view this from the perspective of expectation/unbiasedness. In point of fact, SGD for us is just a computational tool, and we do not utilize random sampling of data within a device. Furthermore, our characterization is in terms of first-order optimality condition, and as such, at any round, we seek an average over all of the data.
>
> Honglin Yuan and Tengyu Ma.  Federated accelerated stochastic gradient descent.arXiv preprintarXiv:2006.08950, 2020.
>
> Zhize Li, Dmitry Kovalev, Xun Qian, and Peter Richtarik.  Acceleration for compressed gradientdescent in distributed and federated optimization.arXiv preprint arXiv:2002.11364, 2020c.

---

> > ### Author Response · Authors · 2020-11-19
> > **Continued**
> >
> > Q2: Missing regularizer in the objective formulation,
> >
> > *A global regularization can be added through device level functions.* We do not have a regularizer in the formulation for simplicity. Since regularizers are data independent, we can add regularizers to the local functions $L_k$. For example, in the case of an L2 regularizer, the global objective becomes $ \arg\min_{\theta \in \mathbb{R}^d} \left [ \frac{\lambda}{2} \|\theta \|^2+\frac{1}{m} \sum_{k \in [m]} L_k(\theta ) \right ] $. We can recover the same global objective as average of local functions by pushing the regularizers to the device level functions as $L_k(\theta ) \leftarrow L_k(\theta ) +\frac{\lambda}{2} \|\theta \|^2$. Therefore, Theorem 1 as well as the convergence rates extend to the regularized objectives. Indeed, in the experiments, we use this idea to prevent overfitting.
> >
> > Q3: Could we also prove convergence for strongly convex functions?
> >
> > *Yes. FedDyn, in fact, achieves linear rate for strongly convex functions which is a significant improvement over convex rate. We refer to the revised draft (see Theorem 1 and Appendix B.2. for analysis.).*
> >
> > Q4: $\alpha > 25L$, tightness of analysis,
> >
> > *The convergence analysis is tight up to constants.* In our analysis, our focus is on the problem dependent variables such as $L,m,P$ so we keep the scalars in a simple integer form. For example, we need both $\kappa$ and $\kappa_0$ to be positive in Lemma 1. This implies that $\alpha \geq (10+\sqrt{140})L$. We used $\alpha > 25L$ to simplify the scalar expression. Indeed, $\alpha$ can be $25L$, we changed it accordingly. Hence, the analysis is tight up to constants.
> >
> > Q5: References; Compression strategy, non IID setting degradation
> >
> > *Compression techniques are orthogonal to our proposed method*. As a future direction, FedDyn can be investigated with a compression technique to further decrease communication costs. We updated our related work.
> >
> > We included papers related to inconsistency of local objectives in our introduction where we talk about this issue.

---

> > > ### Author Response · Authors · 2020-11-19
> > > **Continued**
> > >
> > > Q6: Comparison of FedDyn to proximal splitting algorithms (Condat et al., 2020) including Distributed Davis-Yin (DDY) method,
> > >
> > > We were unaware of this work, and thank the reviewer for pointing us to it. Let us discuss fundamental differences.
> > >
> > > *Partial participation.* Our aim in this work is to deal with  partial device participation (communication), heterogeneity, data imbalance, and massive device levels all together in Federated Learning. Indeed, the proximal splitting algorithms presented in (Condat et al., 2020) including DDY is proposed for the full participation case. Hence, they are not directly suitable for partial communication, which is of a significant concern in Federated Learning.
> > >
> > > *Operational differences.* We first describe DDY algorithm. DDY updates the local devices at each round as
> > > $$\theta^t_k=Prox_{\frac{L_k}{\alpha}}\left(2\theta^{t-1}-s_k^{t-1}-\frac{\nabla L_k(\theta^{t-1})}{\alpha}\right),\ s_k^t=s_k^{t-1}+\theta^t_k-\theta^{t-1}$$
> > > where $\theta^{t-1}$ is the server model, $\theta^t_k$ is the device model and $s_k^t$ is the device state. Devices transmit $s_k^t$s to the server. The server updates its model as, $\theta^t=\frac{1}{m}\sum_{k\in [m]}s_k^t$. The first order condition reveals that $\nabla L_k(\theta^t_k)=- \nabla L_k(\theta^{t-1})+\alpha(2\theta^{t-1}-\theta^t_k-s_k^{t-1})$. On the other hand, in FedDyn, the update relation can be summarized as $\nabla L_k(\theta^t_k)=\nabla L_k(\theta^{t-1}_k)-\alpha(\theta^t_k-\theta^{t-1})$ as described in Eq. 2 (Pg. 4). There are two main differences in these relations as:
> > >
> > > *1. DDY computes $\nabla L_k(\theta^{t-1})$ in each device in each round.* DDY update needs the local gradient at the current server model. In the beginning of each device level optimization, $\nabla L_k(\theta^{t-1})$ needs to be calculated. On the other hand, FedDyn does not require this extra computation.
> > >
> > > *2. FedDyn linearly relates device-wise gradients between consecutive rounds.* This relation significantly simplifies the convergence analysis. Different from FedDyn, DDY linearly relates device-wise gradient of the local model to the device-wise gradient of the server model.
> > >
> > > *Non-Convex Case.* The theory of the proximal splitting algorithms (Condat et al., 2020) is for the convex setting and it is not clear how to carry it to the nonconvex functions. In contrast, FedDyn convergence covers nonconvex functions as stated in Theorem 4. In fact, an important reason our analysis is tractable is because, at the stationary point we have a linear update equation for the gradients of the device models between consecutive rounds. As described earlier, this is not the case for DDY.
> > >
> > > To summarize, there are problem level and operation level differences between FedDyn and the proximal splitting algorithms including DDY.
> > >
> > > Q7: Unnecessary multiplication and division in $h$ update,
> > >
> > > *Yes, we agree that we do not have to multiply and divide by $\alpha$.* However, the current expression helps us to directly see $h=\frac{1}{m}\sum_{k\in[m]}\nabla L_k(\theta_k)$ which makes it easier to reason about convergence.
> > >
> > > Q8: Gradient norm of convergence rate in Theorem 1,
> > >
> > > We added the reference and included a discussion in Appendix B.3 indicating that the current norm arises due to our definition of smoothness with respect to L2 norm and it can be extended to other norms.
> > >
> > > Typos are fixed.

---

> > > > ### Comment · AnonReviewer4 · 2020-11-24
> > > > **My opinion on the authors' response**
> > > >
> > > > I read carefully the response of the authors to the points raised by the different reviewers.
> > > > I find that the authors tend to evade the issues by verbose comments and repeating the same things several times.
> > > > But I take it on the good side and I will keep my score unchanged :  the proposed work is a nice step forward in the brainstorming of the community on how to deal with the communication bottleneck. There are several open and interesting questions, about the links between the proposed method with SGD-type methods on one hand, proximal methods on the other hand, compression, about partial participation... Investigating these questions is left as future work for the authors... and the readers!

---

### Official Review · AnonReviewer2 · 2020-10-28

**Rating:** 7
**Confidence:** 3

**Review:**

Summary:

This paper proposes FedDyn, a dynamic regularization method for federated learning. In FedDyn, the objective function of each active device in each round is dynamically updated, so that the device optimum is asymptotically consistent with the global optimum. The authors consider both the convex and non-convex case, and give convergence rates with theoretical guarantees. In the convex case the proposed algorithm converges faster than the SOTA algorithm SCAFFOLD. In the experiments the authors show that their algorithm takes less communication cost to achieve the same performance than existing algorithms.

The main contribution of the paper includes:

- Proposing the dynamic regularization method to tackle the inconsistency issue in federated learning.

- Proving the convergence rate of the proposed algorithm.

- Saving communication cost compared to existing algorithms, both theoretically and experimentally.


Pros:

- The problem of finding locally consistent distributed algorithms is well motivated. I appreciate the authors' discussion in the introduction.

- The experimental results seem comprehensive. This includes the comparison between FedDyn, SCAFFOLD, FedAvg and FedProx using both synthetic and real datasets in four regimes of interest. The extensive experiments make the claim (saving communication costs) more convincing.

- Overall the paper is well written. The proposed algorithm is well justified with the discussion on the so-called fundamental dilemma. Also the comparison between the proposed algorithm and SCAAFFOLD makes the claim (saving communication costs) much clearer.


Cons:

- What can be said about the computational time of each device during each iteration, compared to that of existing algorithms (say, SCAFFOLD)? This is also mentioned on Page 2 ("This approach, while increasing computation for devices..."). This would be interesting, although I understand the authors focus on a communication point of view.

- The proof techniques of the main theorem somehow seem standard. I did not check other proofs though.

-----------------------------------------

Post-rebuttal:

I appreciate the authors' feedbacks and I keep my evaluation unchanged.

---

> ### Author Response · Authors · 2020-11-12
> **We discuss the computational time aspect of FedDyn, show empirically FedDyn and the baselines have similar computational levels and, although outside the scope of our work, we present a sketch for extending convergence analysis to account for errors in finding stationary points.**
>
> Q: Computational time comparison between FedDyn and the baselines,
>
> Our response has both empirical and computational aspects. We have also included more references related to the communication costs in the submission.
>
> We refer to our answer of R1 (Q2) for empirical justification. We suggest there that FedDyn has similar computation levels compared to the baselines. Furthermore, this computation is adequate to nearly reach stationary solutions in our experiments. Therefore, the need for handling such errors from an experimental perspective lacks strong motivation.
>
> *Theory can be extended to approximate solutions.* The first order condition of device level optimization gives $\nabla L_k(\theta^t_k)=\nabla L_k(\theta^{t-1}_k)-\alpha(\theta^t_k-\theta^{t-1})$ as described in Eq. 2 (Pg. 4). This significantly simplifies our analysis since it linearly relates the gradients of the current round to the previous round. Furthermore, the analysis can be extended to the case where devices do not find stationary points. This error can be modeled as $\nabla L_k(\theta^t_k)=\nabla L_k(\theta^{t-1}_k)-\alpha(\theta^t_k-\theta^{t-1})+e_k^t$ where $e_k^t$s are errors due to using SGD steps in the device level optimization. We split this error into two parts as $e_k^t=\xi_k^t+w_k^t$, where $\xi_k^t$ is the error arising from stochasticity of gradients and $w_k^t$ is the error arising from fixed number of gradient updates.
>
> *1. Error from using fixed number of gradient updates*:
> Gradient descent error decreases with $\frac{1}{K}$ rate where $K$ is the number of gradient descent updates for convex functions. This error will show up as an another term in the convergence rate. We note this term will vanish with higher $K$ values. In the experiments, we see that a similar number of gradient descent updates as with our baselines is sufficient to ignore this effect.
>
> *2. Error from stochasticity of gradients*: Following convention, we can handle SGD errors by noting that $\xi_k^t$s are unbiased estimates of the actual gradient. Now, since our results hold in expectation, most of our lemmas are unaffected. For example, for convex functions, Lemma 2, 3 and 5 will follow from unbiasedness. For Lemma 4 we will need to account for variance, and as it turns out this terms scales as $\frac{1}{\alpha^2}$, but can be handled by including this additional term in our convergence theorem expression.
>
> *Communication vs. Computation Energy Costs.* As stated in Introduction, our focus is communication costs since it is the main source of energy consumption for IoT devices (Yadav \& Yadav, 2016; Latre et al., 2011; Halgamuge et al., 2009). Hence, operational time of IoT devices are mostly limited by the amount of bits communicated rather than the number of local computations. We added these references and pointed out the fact that communication is the main source of energy consumption in the referred paragraph on Pg. 2.
>
> To summarize, empirically FedDyn performs similar levels of computations compared to the baselines. We observe that FedDyn indeed finds nearly stationary points. This motivated us to impose device-wise stationary relation for theoretical analysis. Moreover, the analysis can be extended to handle errors arising from approximately solving device level objectives.
>
> Sarika Yadav and Rama Shankar Yadav. A review on energy efficient protocols in wireless sensor networks. Wireless Networks, 22(1):335–350, 2016.
>
> Benoit Latre, Bart Braem, Ingrid Moerman, Chris Blondia, and Piet Demeester. A survey on wireless body area networks. Wireless networks, 17(1):1–18, 2011.

---

### Official Review · AnonReviewer1 · 2020-10-28
**New federated learning methods for heterogeneous devices**

**Rating:** 7
**Confidence:** 3

**Review:**

This paper proposed a new federated learning algorithm called FedDyn, which was motivated by the observation that the local objectives for each device might lead to inconsistent models between devices for heterogeneous data. Such observation is already observed in SCAFFOLD, and this paper further improves over SCAFFOLD with better communication efficiency. Both theoretical convergence rates and empirical  results about communication efficiency are presented to support the advantages of FedDyn methods.

I have the following comments on the paper:
1. The choice of alpha: it seems alpha is an important hyperparameter which can largely affects the theoretical convergence and empirical efficiency of the proposed FedDyn method. Unfortunately, I could not find much discussion in the paper about alpha: I think how alpha affect the convergence rates, and how alpha is chosen in the empirical results should be discussed, it would be better if the authors could provide an empirical study about the parameter sensitivity in alpha.

2. How the local objective is optimized: since at each round FedDyn requires each device to solve a local optimization problem which would requires iterative algorithms (such as SGD) to find an approximate solution, how to solve the local optimization problem, and how the accuracy of the local optimization affects theoretical communication efficiency as well as empirical results should be discussed.

3. Related work: there is a related paper (to appear in NeurIPS 2020) which proposes to solve a similar local objective, it would be good to discuss the differences: FedSplit: an algorithmic framework for fast federated optimization https://arxiv.org/abs/2005.05238.

---

> ### Author Response · Authors · 2020-11-12
> **In experiments FedDyn computation is identical with respect to the prior works. Outline differences between reviewer cited FedSplit and ours; We describe the effect of $\alpha$ parameter.**
>
> Q1: Sensitivity analysis of $\alpha$,
>
> Our response has both theoretical and empirical aspects.
>
> *In theory, $\alpha$ has no effect on asymptotic convergence rate.* $\alpha$ is an important parameter of FedDyn. $\alpha$ balances two problem dependent constants as shown in Theorem 2, 3, and 4 for convex, strongly convex and nonconvex functions respectively. Consequently, optimal value of $\alpha$ depends on these constants. Since all of these constants are independent of $T$, the value of $\alpha$ is independent of $T$. Hence, $\alpha$ does not impact the dependence on $T$ in the convergence rate.
>
> *In Experiments $\alpha$ is a hyperparameter.* We use hyperparameter search to find the best $\alpha$ value as stated in Appendix A.2. In passing, we point out that almost all Federated Learning including fedavg, fedprox and scaffold require hyperparameter tuning. As to how to bypass hyperparameter tuning is important but is outside the scope of our work.
>
> To summarize, $\alpha$ is the main parameter of FedDyn. However, $\alpha$ value does not impact dependency on $T$ in the convergence rate expression. These results, as well as an additional empirical sensitivity analysis using different $\alpha$ values are presented in Appendix A.3.
>
> Q2: How does FedDyn optimally solve device level problem?
>
> Our response will consider both empirical and theoretical aspects. For the theoretical issues pertaining to handling this issue we refer to our Reviewer 2 response.
>
> *In experiments we implement FedDyn to match the same amount of computation as prior works.* For example, FedDyn and SCAFFOLD propose the same number of epochs as us for Cifar10 and Cifar100 experiments as indicated in Appendix A.2 hence their local computation levels are identical. As shown in Tables 1-6, FedDyn still outperforms the baselines despite performing similar computations.
>
> *In experiments FedDyn finds near stationary solutions.* For instance, in Cifar10, IID, 100 devices, 10\% participation setting. We have observed that the average gradient norm of device models in FedDyn is vanishingly small relative to other variables (less than the 4-th decimal place). As such from a practical perspective, the fact that we do not solve to optimality does not appear to be practically important. This justifies our claim that the given amount of computation with SGD updates for FedDyn (instead of a solver) is sufficiently large that the updated models are essentially stationary points.
>
> To summarize, empirically FedDyn performs similar levels of computations compared to the baselines. We observe that FedDyn indeed finds near stationary points. This motivated us to go with device-wise stationary relation for the theoretical analysis. Moreover, the analysis can be extended to handle errors arising from approximate solutions.

---

> > ### Author Response · Authors · 2020-11-17
> > **Continued**
> >
> > Q3: Differences between FedDyn and FedSplit,
> >
> > We were unaware of this work, and thank the reviewer for pointing us to it. Let us outline a few major differences.
> >
> > *Scope.* As described in our introduction, our goal with FedDyn is to be able to handle partial device participation (communication), heterogeneity, data imbalance, and massive device levels all together. As such proposed FedSplit does not account for partial communication, which is a significant issue in the context of federated learning.
> >
> > *FedSplit evaluation with partial participation.* During this rebuttal period, we implemented FedSplit with partial device participation (see Appendix A.4). To do so we experimented with different modifications of FedSplit that appear reasonable (nevertheless we do not suggest our modifications are optimal). On CIFAR10, 100 devices, $10$% participation setting, these adapted versions of FedSplit perform significantly worse than FedDyn and SCAFFOLD.
> >
> > *Conceptual Difference.* We describe differences for full participation. FedSplit update per device per round is $z_j^{t+\frac{1}{2}}=Prox_{sf_j}(2x^t-z_j^t)$ and $ z_j^{t+1}=z_j^t+2(z_j^{t+\frac{1}{2}}-x^t)$ where devices transmit $z_j^t$s and keep $z_j^{t+\frac{1}{2}}$s as an internal state. The server averages device models and sets the server model as $x^t=\frac{1}{m}\sum_{i\in[m]}z_i^t$. The first order condition of device level optimization gives $s\nabla f_j(z_j^{t+\frac{1}{2}})+z_j^{t+\frac{1}{2}}-2x^t+z_j^t=0$. So, the relation of device-wise gradients ($\nabla f_j(z_j^{t})$) between consecutive rounds is implicit. On the other hand, in FedDyn, the first order condition gives  $\nabla L_k(\theta^t_k)=\nabla L_k(\theta^{t-1}_k)-\alpha(\theta^t_k-\theta^{t-1})$ as described in Eq. 2 (Pg. 4). This explicitly, in fact, linearly relates device-wise gradients between consecutive rounds and significantly simplifies the convergence reasoning (see the paragraph starting with 'Key Property of Algorithm' on pg. 4). Hence, the update mechanisms of FedDyn and FedSplit are conceptually different.
> >
> > *Empirical FedSplit comparison for full participation.* Our FedSplit discussion in Appendix A.4 includes an experiment on Cifar10 with full participation. We observe that FedDyn outperforms FedSplit even in full participation level.
> >
> > *Non-Convex Case.* As it stands, FedSplit theory is applicable to the convex setting, and no analysis for non-convex case is presented. On the other hand, our results hold for nonconvex functions as stated in Theorem 4. Indeed, an important reason our analysis is tractable is because, at the stationary point we have a linear update equation for the gradients between consecutive rounds. This is no longer the case for FedSplit since the gradients between consecutive rounds are implicitly related as we described earlier.
> >
> > To summarize, FedDyn and FedSplit exhibit differences in terms of scope and operation. Empirical results reveals Scaffold and FedDyn outperform FedSplit for both full and partial participation levels (see Appendix A.4).

---

### Comment · ~Mingyi_Hong1 · 2021-02-21
**Relation with FedPD Algorithm**

Thank you for the very nice work, and the clear intuition provided for the dynamic regularization.

I have a question regarding the relationship between the proposed algorithm and the FedPD algorithm. In the paper, it is  commented that the difference between these two algorithms are mainly in the way that they save communications. But I wonder if, both of the algorithms do not attempt to save communication (i.e., FedPD does not skip communication, while the proposed algorithm use full user participation), then what's the relationship between them?

We have some preliminary derivations, see the link below. It will be great that we can discuss about this issue. Thank you!

http://people.ece.umn.edu/~mhong/FedDyn_FedPD.pdf

---

> ### Comment · ~Narsil_Zhang1 · 2021-03-01
> **The equivalence holds in math.**
>
> The note directly shows the equivalence of FedPD and FedDyn (there might still be some difference in specific implementation). As a result, I think that the authors should at least refer to this note in a revision of camera ready version for a fair development of federate community.

---

> ### Author Response · Authors · 2021-03-01
> **Comparison of FedDyn and FedPD**
>
> Thanks for interest in our paper. While we agree that FedPD and FedDyn are related in the $100\\%$ device participation case. As such this relationship appears to suggest that FedDyn can be viewed from a primal dual perspective. Nevertheless, this relationship is just coincidence, and once we consider partial participation, we can no longer view FedDyn algorithm as a variant of primal-dual method. Before, pointing this out, observe that assumptions underlying FedPD is somewhat impractical. Namely, FedPD assumes either all devices at once communicate to a server, or no device communicates. This is clearly unrealistic, when one considers massive distributed device scenarios, where devices asynchronously participate based on their local properties (i.e., reliable communication link, battery capacity etc.), and as such communications to server across devices is random. For the server to wait to update until all devices respond, will significantly increase latency. In contrast, FedDyn allows for completely asynchronous updates.
>
> Now coming to the more substantive point, namely, how would a primal-dual extension look like in the partial participation case? For this we refer to previous distributed optimization papers. There two aspects are fundamentally different: (a) Technically, finite delay from each device, which is independent of the number of devices is enforced. This assumption does not hold with random partial participation; (b) The algorithm requires averaging across both stale and current models in order to ensure convergence; Empirically, we have observed, uniformly across all of the datasets, that performance of such a scheme is significantly worse than conventional federated learning.
>
> In contrast, FedDyn updates based on current active devices, and guaranteeing convergence requires steps parallel to SCAFFOLD. Conceptually, FedDyn is based on noting that $L_k(\theta)$ is biased with respect to the global objective, and the linear approximation, $\langle \nabla L_k(\theta^t_k),\theta \rangle$ is a means to mitigate this bias. Indeed, we chose one natural variant, but there are several variants, such as say, $\langle L_k( \theta_t), \theta\rangle$, and many others that serves equally well. These variants, in particular, bear little relationship with primal-dual algorithms even in the full-participation case. Identifying which one of these variants result in good performance is the subject of our future work.

---

> > ### Comment · ~Ming_Yan1 · 2021-04-14
> > **Connection to ADMM**
> >
> > It is true that the partial participation is very interesting in FL.
> > I just want to also mention the connection to distributed ADMM for the full participation case.  Introduce a variable $\theta_0$ at the server, and let $\theta_0=\theta_k$ be the constraint. In this case $h^t$ at the server will be the average of the gradient $\nabla L_k(\theta_k^t)$. The update of $\theta_0$ is the update of $\theta^t$. The gradient in the algorithm is the dual variable (or multiplier).
> >
> > For the partial participation case, it is not equivalent to ADMM, and there are many challenges in this case. Thanks for the nice work.

---

### Comment · ~Sai_Praneeth_Karimireddy1 · 2021-04-12
**Mistake in Algorithm and comparision to SCAFFOLD**

Dear FedDyn authors,

Thank you for the very nice work. Reducing the per-round communication requirement of SCAFFOLD by moving the global $\mathbf{c}$ correction to the server was very innovative!

A few comments:

1. There seems to be a **mistake in Algorithm 1** as is currently written. Consider the following 1D problem with 2 devices both of which participate every round:
$$
L_1(\theta) = \theta \quad \quad \text{and} \quad \quad  L_2(\theta) = -2\theta ,\quad  \text{ for } \theta \in [0,1].
$$
Over the range $\theta \in [0,1]$, the minima is at 1. However, if we run Algorithm 1 starting from $\theta^0 = 0$, it will always remain at 0 with $\theta^t_k = 0$ for all $t$ and $k$. I believe this is because the **initialization is incorrect**  and $\nabla L_k(\theta^0_k)$ cannot be set to 0 as is currently done (perhaps this was a typo?).


2. Since the clients compute full gradients, the **variance is 0**. In this setting, SCAFFOLD actually obtains a rate of i) $O(\sqrt{\frac{m}{P}} \frac{1}{T})$ for general convex functions, and ii) $O(\sqrt[2/3]{\frac{m}{P}} \frac{1}{T})$ for non-convex functions. FedDyn matches the rate for general convex functions, but for non-convex functions **FedDyn is slower than SCAFFOLD**.  In an earlier version of our paper, our analysis for the general-convex setting was slightly loose since we did not focus on the case when $\sigma = 0$. Please see *Remark 11* in our [latest version](https://arxiv.org/abs/1910.06378).

Thanks!

SCAFFOLD authors

---

> ### Author Response · Authors · 2021-04-12
> **Algorithm Clarification**
>
> Thanks for your kind comments and interesting question about initialization. As such the initialization is fine. However, the algorithm has to be modified to account for constrained optimization. We can do it in two ways: (a) express it as an unconstrained problem by using multipliers; (b) Solve an unconstrained problem at the devices, and during server update perform a model projection step. This model projection step is to ensure that we get a solution in the feasible set. For instance for the specific example we would project the parameter $\theta$  into the interval [0,1] at every round. In general, this can be done by projecting the server averaging $\theta^t=\left(\frac{1}{|P_t|} \sum_{k \in {\cal P}_t} \theta_k^t\right)-\frac{1}{\alpha}h^{t}$ (Algorithm 1) onto the global constraint set.

---

> > ### Comment · ~Sai_Praneeth_Karimireddy1 · 2021-04-15
> > **Problem remains for unconstrained**
> >
> > Thanks for the fast response. We could run the algorithm by dropping the constraints, (or using projection steps are mentioned in the answer). Either way, the problem remains.
> >
> > Consider the above example where $L_1(\theta) = \theta$ and $L_2(\theta) = -2\theta$.
> > The client gradients can be computed as $\nabla L_1(\theta) = 1$ and $\nabla L_2(\theta) = -2$  and are both constant. We thus have for all $k$ that
> > $$
> > L_k(\theta) - \langle\nabla L_k(\theta) , \theta \rangle  = 0 .
> > $$
> >
> > This means that the subproblem solved by FedDyn is
> > $$
> > \theta^t_k = \text{arg}min_{\theta} L_k(\theta) - \langle\nabla L_k(\theta) , \theta \rangle  + \tfrac{\alpha}{2}\|| \theta - \theta^{t-1} \||^2 = \text{arg}min_{\theta} \tfrac{\alpha}{2}\|| \theta - \theta^{t-1} \||^2  = \theta^{t-1}\,.
> > $$
> > Thus, throughout we will have $\theta^t_{k-1} = \theta^0 = 0$. The algorithm makes no updates and remains stuck at initialization.
> >
> > However, clearly the global function is $-\tfrac{1}{2}\theta$. To minimize this function, we need to move towards larger positive numbers.

---

> > > ### Author Response · Authors · 2021-04-15
> > > **Let us actually work out the example with FedDyn**
> > >
> > > Perhaps the brevity in our response was not sufficiently clear, and we apologize in advance for the confusion. However, as observed by scaffold authors in the first message, note that our initialization for all the states and parameters are zero (see Alg. 1). The reason this initialization makes sense is that, otherwise, the states would need to be communicated initially to the server. With this in mind let us work out the example posed.
> > >
> > > Assume $\alpha=10$, and as described above and as well as pointed out in our algorithm, start with $\theta^0=\nabla L_1(\theta_1^0) = \nabla L_2(\theta_2^0) = h^0=0$.
> > >
> > > In round 1: $\theta_1^1 = -0.1$ and $\theta_2^1 = 0.2$, and local states $\nabla L_1(\theta_1^1)=1$ and $\nabla L_2(\theta_2^1)=-2$. On the server side, $h^1=-0.5$ and $\theta^1=0.1$. This can be verified by our equations, namely,
> > >
> > > $\theta_k^t=\arg \min_{\theta} L_k(\theta) - \langle \nabla L(\theta_k^{t-1}),\theta \rangle + \frac{\alpha}{2}\|\theta - \theta^{t-1}\|^2.$
> > >
> > > Let us work this out for the provided example to avoid any misunderstanding:
> > >
> > > $\theta_1^1=\mbox{arg} \min_{\theta} \theta - \langle \nabla L(\theta_1^0),\theta \rangle + \frac{\alpha}{2}\|\theta - \theta^0\|^2 = \mbox{arg} \min_{\theta} \theta - 0 + 5 \theta^2 \implies \theta_1^1 = -0.1$
> > >
> > >  Next, consider update for the second device:
> > >
> > > $\theta_2^1=\mbox{arg} \min_{\theta} -2\theta - \langle \nabla L(\theta_2^0),\theta \rangle + \frac{\alpha}{2}\|\theta - \theta^0\|^2 = \mbox{arg} \min_{\theta} -2\theta - 0 + 5 \theta^2 \implies \theta_2^1 = 0.2.$
> > >
> > > Finally, on the server side, we get
> > >
> > > $h^t = h^{t-1} - \alpha \frac{1}{m} \left(\sum_{j=1}^2 \theta_j^t-\theta^{t-1}\right),\ \ \ \theta^t=\frac{1}{m}\left(\sum_{j=1}^2 \theta_j^t\right) - \frac{1}{\alpha}h^t$
> > >
> > > Plugging in it is easily verified that, $h^1=-0.5$ and $\theta^1=0.1$. Now let us go to round 2: observe that $\theta_1^2 = \theta_2^2=0.1,\ \ \ \nabla L_1(\theta_1^2)=1,\ \ \  \nabla L_2(\theta_2^2)=-2$. As for the server we have $h^2=-0.5$. The server model is now updated to $\theta^2 = \frac{1}{2}(\theta_1^2 + \theta_2^2) - \frac{1}{\alpha}h^2 = 0.1 + 0.05 = 0.15$. For round 3, a further increment follows in the server model resulting ultimately in convergence to the optimal solution.
> > >
> > >
> > > We provide a plot of the trajectory with the number of rounds appears in this [link](https://github.com/alpemreacar/FedDynClarification/blob/main/toy_example.png).

---

> ### Author Response · Authors · 2021-04-19
> **Convergence Rate Clarification**
>
> Thanks for your kind comments and the remark on the convergence rates.
>
> For the non-convex setting, Theorem 7 in SCAFFOLD builds upon the so called control variate for each round, $\Xi_r$,  (Lemma 17 pg 32), which we were unable to verify. Please see [Lemma17_SCAFFOLD](https://github.com/alpemreacar/FedDynClarification/blob/main/Lemma17_SCAFFOLD.pdf). As such the non-convex rate derived in Theorem 7 is based on Lemma 17.

---

### Decision · Program_Chairs · 2021-01-08
**Final Decision**

**Decision:**

Accept (Oral)

**Comment:**

The paper introduces a new federated learning algorithm that ensures that the objective function optimized on each device is asymptotically consistent with the global loss function.    Both theoretical analysis and empirical results, evaluating communication efficiency, demonstrate the advantages of the proposed FedDyn method over the baselines.

All the reviewers recommend accepting the paper. To summarize the discussion:

- R1 mentioned a very recent (NeurIPS 20) related paper and asks several questions. I believe that the authors nicely answered the questions and discussed the relation to the previous paper in detail.

-  R2 mentioned that the paper focuses solely on minimizing communication costs, ignoring costs of local computations. The authors argued that the local computation costs are comparable to those of the baselines, and, in general, communication costs are the main source of computation energy costs (pointing to previous work), and, thus, are a natural objective to optimize.  I believe that this adequately addressed this (and other) reviewer's concerns and the reviewer kept their score unchanged.

- R3 had several concerns, which according to the reviewer were addressed in the rebuttal (they increased the score).

- R4 points out several limitations of the method and theoretical analysis and believes that the rebuttal did not quite address the concerns. Nevertheless, remains positive about the paper, and believes that the shortcomings can be addressed in follow-up work.

We share the reviewers' sentiment: it is a very nice and interesting paper, and should be accepted.